# 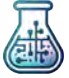 `ProBio`: A Protocol-guided Multimodal Dataset for Molecular Biology Lab

**Jieming Cui** [1,2,★]
cuijieming@stu.pku.edu.cn

**Ziren Gong** [3,★]
ziren.gong@outlook.com

**Baoxiong Jia** [2,★]
jiabaoxiong@bigai.ai

**Siyuan Huang** [2]
syhuang@bigai.ai

**Zilong Zheng** [2,✉]
zlzheng@bigai.ai

**Jianzhu Ma** [3,4✉]
majianzhu@tsinghua.edu.cn

**Yixin Zhu** [1,2,5,✉]
yixin.zhu@pku.edu.cn

[★] J. Cui, Z. Gong, and B. Jia contributed equally.    [✉] corresponding authors
[1] Institute for Artificial Intelligence, Peking University
[2] National Key Laboratory of General Artificial Intelligence
[3] Institute for AI Industry Research, Tsinghua University
[4] Department of Electronic Engineering, Tsinghua University
[5] PKU-WUHAN Institute for Artificial Intelligence

**https://probio-dataset.github.io**

## Abstract

The challenge of replicating research results has posed a significant impediment to the field of molecular biology. The advent of modern intelligent systems has led to notable progress in various domains. Consequently, we embarked on an investigation of intelligent monitoring systems as a means of tackling the issue of the reproducibility crisis. Specifically, we first curate a comprehensive multimodal dataset, named 🧪 **ProBio**, as an initial step towards this objective. This dataset comprises fine-grained hierarchical annotations intended for studying activity understanding in Molecular Biology Lab (BioLab). Next, we devise two challenging benchmarks, transparent solution tracking, and multimodal action recognition, to emphasize the unique characteristics and difficulties associated with activity understanding in BioLab settings. Finally, we provide a thorough experimental evaluation of contemporary video understanding models and highlight their limitations in this specialized domain to identify potential avenues for future research. We hope 🧪 **ProBio** with associated benchmarks may garner increased focus on modern AI techniques in the realm of molecular biology.

## 1   Introduction

Despite notable progress in scientific research, the challenge of reproducing research findings has surfaced as a significant obstacle. Baker (2016) suggests that a significant proportion of researchers, exceeding 70%, have reported unsuccessful attempts to replicate experiments carried out by their colleagues, primarily attributed to inadequate clarity of protocols (Ioannidis, 2005; Begley and Ellis,

37th Conference on Neural Information Processing Systems (NeurIPS 2023) Track on Datasets and Benchmarks.

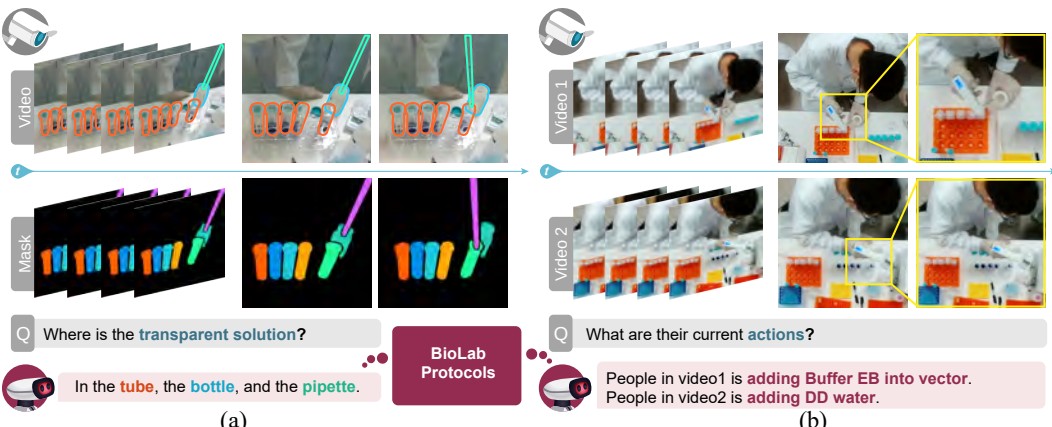

Figure 1: **An overview of two challenging tasks identified and presented in 🧪 ProBio.** 🔵 denotes a set of cameras, and 🔴 denotes intelligent monitoring models with access to BioLab protocols. Task (a): track transparent solutions. Task (b): action understanding guided by protocols. Object IDs share the same color across frames and in the texts.

2012). These protocols (*i.e.*, procedural instructions) frequently exclude crucial factors, such as temperature or pH, that can substantially impact the results. As a result, researchers rely heavily on mentorship from seasoned experts when conducting experiments. The management of such experiments necessitates a significant investment of time and resources, often unattainable for many laboratories, particularly in Molecular Biology Labs (BioLabs), wherein replicating experiments is more time-consuming and costly (Calne, 2016). Since modern intelligent systems have brought significant advancements in various fields (Grosan and Abraham, 2011; Gretzel, 2011; Stephanopoulos and Han, 1996; Tavakoli et al., 2020), there is a growing need to develop an AI assistant to tackle this reproducibility crisis. Toward building such an AI assistant, we set off to curate a multimodal dataset recorded in BioLabs with benchmarks of modern AI methods.

Due to the inherent characteristics of BioLab, constructing this multimodal dataset faces **two grand challenges**. The **first** one is the lack of readily available protocols with *sufficient* details; existing ones typically only provide high-level guidance (Latour, 1987; Cetina, 1999; López-Rubio and Ratti, 2021; Peterson and Panofsky, 2021), lacking details necessary for reproducing results step by step. An example of process failure in cell culturing occurs when the culture medium is not inverted correctly after the addition of yeast. Nevertheless, such crucial information is frequently regarded as a standard practice in research and disregarded in written protocols. The heterogeneity of experimental instructions exacerbates the complexity of this scenario, as different labs document identical tasks in diverse manners contingent upon their respective resource availability (Braybrook, 2017). Therefore, curating standardized protocols with *sufficient* details, coupled with videos of each instruction's execution, is essential for building an AI assistant for BioLab.

The **second** challenge pertains to comprehending domain-specific actions and objects at the intricate level of granularity. From the computer vision perspective, this poses a challenging task for achieving fine-grained understanding in contrast to typical scenarios like sports (Shao et al., 2020; Xu et al., 2022) or instructional videos (Zhang et al., 2023; Tang et al., 2019; Miech et al., 2019; Das et al., 2013; Zhou et al., 2018; Zhukov et al., 2019). The complexity of event understanding in BioLab is primarily attributed to the specialized instruments and the ambiguity of actions involved. For instance, experiments commonly involve liquid transfer between visually similar and transparent containers (Liang et al., 2016, 2018), posing additional challenges in object detection and event parsing (Jia et al., 2020; Huang et al., 2023). Moreover, actions that are perceptually similar may have divergent semantic meanings across various experiments due to the strong dependence between actions and experimental contexts (Stacy et al., 2022; Jiang et al., 2022, 2021; Chen et al., 2021). These visual ambiguities (Fan et al., 2022; Zhu et al., 2020; Zhu, 2018) have been mostly left untouched in prior arts (Murray et al., 2012; Shao et al., 2020; Goyal et al., 2017; Kay et al., 2017; Zhu et al., 2022; Panda et al., 2017; Kanehira et al., 2018) and present an ideal and unique testbed for *multimodal* video understanding (Wang et al., 2022b; Huang et al., 2023).

We present 🧪 ProBio, the first protocol-guided multimodal dataset in BioLab to tackle the above challenges. 🧪 ProBio provides (i) a meticulously curated set of detailed and standardized protocols

Table 1: **A comparison between 🧪 ProBio and existing activity understanding datasets.** We use **segments** to denote the number of object segmentation maps annotated, **activity.cls** to denote the number of protocols, **activity.num** to denote the number of clips that align with protocols.

| Domain | Dataset | Duration | Segments | Procedure | Instruction | HOI pairs | Multi-view | Hierarchy | Activity cls | Activity num |
|--------|---------|----------|----------|-----------|-------------|-----------|------------|-----------|-----|-----|
| kitchen | EPIC-Kitchen (2020) | 100h | 90,000 | ✓ | ✓ | ✗ | ✗ | ✗ | 20,000 | 39,596 |
| | YouCook2 (2018) | 175.6h | 13,829 | ✓ | ✓ | ✗ | ✗ | ✗ | 89 | 2,000 |
| | LEMMA (2020) | 10.1h | 11,781 | ✗ | ✓ | ✗ | ✓ | ✓ | 15 | 324 |
| daily | COIN (2019) | 476.63h | 46,354 | ✓ | ✓ | ✗ | ✗ | ✓ | 180 | 11,287 |
| | HowTo100M (2019) | 134,472h | 136M | ✗ | ✓ | ✗ | ✗ | ✗ | 12 | 23,611 |
| | HOMAGE (2021) | 25.4h | 1752 | ✗ | ✓ | ✓ | ✓ | ✓ | 453 | 24,600 |
| | IAW (2023) | 183h | 48,850 | ✓ | ✓ | ✗ | ✓ | ✗ | 14 | 420 |
| sport | FineGYM (2020) | 708h | - | ✓ | ✓ | ✗ | ✓ | ✓ | 15 | 4,883 |
| | FineDiving (2022) | 57.9h | - | ✓ | ✓ | ✗ | ✓ | ✓ | 52 | 3,000 |
| BioLab | 🧪 **ProBio** | 180.6h | 213,361 | ✓ | ✓ | ✓ | ✓ | ✓ | 79 | 3,724 |

with corresponding video recordings for each experiment and (ii) a natural and systematic evaluation framework for fine-grained multimodal activity understanding; see Fig. 1. We construct 🧪 ProBio by selecting a set of 13 frequently conducted experiments and augmenting existing protocols by incorporating three-level hierarchical annotations. This configuration yields 3,724 practical-experiment instructions and 37,537 Human-Object Interaction (HOI) annotations with an overall length of 180.6 hours; see Sec. 3. We design two tasks in 🧪 ProBio: transparent solution tracking and multimodal action recognition, assessing models' capability to leverage both visual observations and protocols to discern unique environmental states and actions. In light of the significant disparities observed between human and model performance in action recognition, we devise diagnostic splits that stratify experimental instruction into three categories based on difficulties (*i.e.*, easy, medium, hard). We hope 🧪 ProBio and associated benchmarks will foster new insights to mitigate the reproducibility crisis in BioLab and promote fine-grained multimodal video understanding in computer vision.

This paper makes three primary contributions:

- We introduce 🧪 ProBio, the first protocol-guided dataset with dense hierarchical annotations in BioLab to facilitate the standardization of protocols and the development of intelligent monitoring systems for reducing the reproducibility crisis.

- We propose two challenging benchmarking tasks to measure models' capability in leveraging both visual observations and language protocols for fine-grained multimodal video understanding, especially for ambiguous actions and environment states.

- We provide an extensive experimental analysis of the proposed tasks to highlight the limitations of existing multimodal video understanding models and point out future research directions.

## 2 Related work

**Fine-grained activity datasets**    Action understanding has been a long-standing problem in computer vision with successful attempts in data curation (Murray et al., 2012; Kay et al., 2017; Soomro et al., 2012; Caba Heilbron et al., 2015; Monfort et al., 2019). To provide fine-grained activity annotations, datasets (Goyal et al., 2017; Stein and McKenna, 2013; Damen et al., 2020; Jia et al., 2020; Rai et al., 2021; Luo et al., 2022b; Grauman et al., 2022) come with HOI labels, object bounding boxes, hand masks, *etc*. Tab. 1 compares 🧪 ProBio with existing datasets.

The task of delineating intricate action hierarchies for daily activities is challenging. One line of work justifies action hierarchy design by examining activities in sports (Shao et al., 2020; Xu et al., 2022) and kitchens (Kuehne et al., 2014; Li et al., 2018; Damen et al., 2020). The annotations provided, while detailed, may lack strong contextual information and may not be suitable for complex tasks that demand nuanced multimodal understanding. Another line of work leverages furniture assembly (Ben-Shabat et al., 2021; Sener et al., 2022; Zhang et al., 2023) as a means to highlight action dependencies. Nonetheless, the practical applications of these tasks are limited.

**Multimodal video understanding**    Complex video understanding tasks require leveraging context in addition to direct visual inputs. Instructional videos (Zhou et al., 2018; Tang et al., 2019; Zhukov et al., 2019; Miech et al., 2019), as the most readily available multimodal video learning source, have been frequently utilized for various multimodal video understanding tasks, such as retrieval

(Anne Hendricks et al., 2017; Wang et al., 2019), captioning (Zhou et al., 2018; Xu et al., 2016; Yu et al., 2019), and question answering (Li et al., 2016; Lei et al., 2018; Grunde-McLaughlin et al., 2021; Xiao et al., 2021; Yang et al., 2021; Jia et al., 2022). These datasets are oftentimes large in scale and curated from internet videos with language primarily sourced from online encyclopedia platforms or transcribed from subtitles. The quality of the language modality is considerably impeded by the substantial human effort required to refine insufficient and inaccurate language descriptions (Miech et al., 2019). In addition, the extensive scale of data necessitates the frequent utilization of pre-trained models from other modalities, such as images, for the purpose of multimodal video understanding (Wang et al., 2021; Zellers et al., 2021; Xu et al., 2021; Luo et al., 2022a). Nevertheless, adapting such pre-trained models to specialized domains, such as BioLabs, presents a formidable challenge due to the distinctive nature of objects and actions involved. To tackle these issues, 🧪 **ProBio** provides aligned video-protocol pairs, accompanied by detailed experimental instructions for every procedure. Benchmarks on 🧪 **ProBio** offer a comprehensive examination of existing models for multimodal video comprehension in specific domains.

## 3 The 🧪 **ProBio** Dataset

🧪 **ProBio** consists of 180.6 hours of multi-view recordings that encompass 13 common biology experiments conducted in BioLab. This section introduces the collection (Sec. 3.1) and annotation (Sec. 3.2) process of 🧪 **ProBio** and describes the associated benchmarking tasks (Sec. 3.3).

### 3.1 Data collection

**Biology protocol** We have assembled a collection of standard protocols along with comprehensive instructions, and videos guided by these protocols, all included in 🧪 **ProBio**. We collect our protocol database by first crawling publicly available protocols published in top-tier journals and conferences. As these protocols often contain only high-level instructions, commonly referred to as brief experiments (`brf_exp`), we construct an online annotation tool for seasoned researchers to augment them with additional experimental instructions, commonly referred to as practical experiments (`prc_exp`); of note, this augmentation could be different from experiments to experiments, resulting in a one-to-many mapping from brief to practical experiments. After this augmentation, we select 13 brief experiments with multiple practical experiments and instruct seasoned researchers in BioLab to perform. Please refer to Appx. A.2 for additional details.

**Monitoring video** The video data is recorded in a laboratory that adheres to the international standard for molecular biology (Nest.Bio Labs, 2023). A total of ten cameras are installed to oversee all experimental procedures. To ensure the quality and clarity of the collected videos, we consult with experienced biological researchers regarding the cameras' viewpoints and positions. A total of eight high-resolution RGB cameras are affixed above the operation tables and instruments. To capture intricate HOIs in detail, two supplementary RGB-D cameras have been positioned in close proximity to the primary operating table and sterility chamber. Over 700 hours of video are collected through a 24-hour monitoring process, which minimizes disruption to the researchers' regular activities. The raw videos undergo additional processing through two steps: (i) automatically filtering of no-action frames using OpenPose (Cao et al., 2017) and YOLOv5 (Ultralytics, 2022), and (ii) manual removal of frames depicting actions unrelated to the intended focus, such as conversing, note-taking, or texting. A total of 180.6 hours hours video pertaining to the 13 selected brief experiments has been obtained.

### 3.2 Data annotation

In molecular biology experiments, it is common for routine operations to occur periodically. To facilitate the annotation process, a representative and distinct subset of video clips is selected. This subset consists of a total of 9.64 hours top-down view videos and 1.05 hours nearby-view videos, marked with detailed action labels to provide clear, fine-grained information. During the process of annotation, we consider (i) detailed HOIs in the form of HOI pairs for each frame and (ii) object segmentation masks for each interacted object. To establish a connection between the fine-grained annotations and the underlying biological experiment, supplementary annotations are furnished to denote the precise location of each action within the brief and practical experiments. This process establishes a hierarchical structure consisting of three levels, encompassing fine-grained action data for multimodal video understanding and categorical information for future research on intelligent

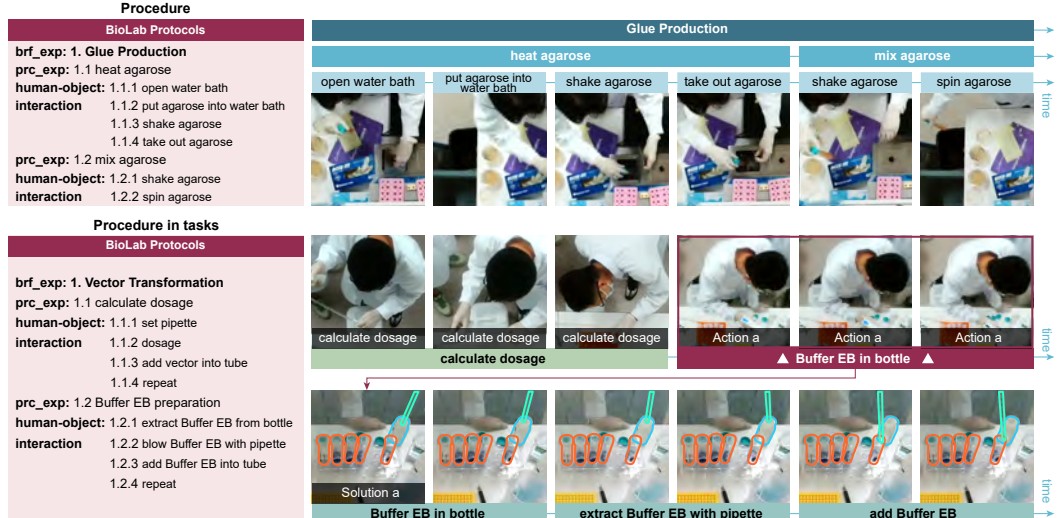

Figure 2: **An example of BioLab protocols, data, and tasks.** We show activities recorded (right) and their corresponding protocols (left). HOI annotations are visualized in the top row. The bottom row gives an example of how knowledge in protocols guides (i) the recognition of actions (in **red**) given matched actions (green) and (ii) tracking the transparent solution status (blue).

monitoring systems. When exporting annotations, we translate them to a list of indexes to collect the relations between humans and objects (*e.g.*, [["human_1," "object_2"], "inject"] indicates "human_1" and "object_2" has a relation of type "inject"). To improve the precision of our annotations, we organized our dataset into 16 separate batches, with each batch comprising between 1,000 and 5,000 frames. After the annotation group finished labeling each batch, we engaged in 2 to 3 rounds of thorough reviews with biological experts. These specialists helped us detect and rectify any errors in the labels and their corresponding relationships. Any sections that were identified as inaccurate underwent a process of correction and re-annotation.

We annotate a list of 48 objects and 21 action verbs in HOIs based on their significance confirmed by seasoned biology researchers. A team of crowd-sourced labeling workers was recruited to annotate the selected videos, following appropriate training in the annotation process. In total, we obtained 37,537 HOI annotations and 213,361 object segmentation maps for all object categories. To further enhance comprehension of alterations of object status, supplementary labels for object status were furnished over segmentation maps in the nearby-view video recordings. As transparent solutions and containers are commonly employed in molecular biology experiments, additional solution annotations primarily pertain to the transparent solution types inside containers such as tubes or pipettes, resulting in additional 40,443 additional labels (*e.g.*, ["tube_1," "LB_solution"] for test tube with LB_solution). Fig. 2 shows an example of annotations. The task of annotation entails establishing a correspondence between the present state of videos and practical experiments (*i.e.*, prc_exp) through the allocation of action labels, thereby enabling the subsequent annotation of more detailed actions. Please refer to Appx. A.2 for additional details on the annotation process.

### 3.3  Benchmark design

We devise two benchmarking tasks associated with 🧪 **ProBio**, aimed at enhancing multimodal video understanding. These two tasks are referred to as transparent solution status tracking (TransST) and multimodal action recognition (MultiAR). We present the statistics of data and annotations utilized in each task in Tab. 2 and explicate the settings of each task as outlined below.

**Transparent solution tracking (TransST)**    Tracking and understanding object status in BioLabs is challenging due to their visual ambiguity as visualized in Fig. 2. Multiple factors contribute to this challenge: (i) most objects in biology experiments are small and difficult to detect and track, (ii) containment relationships obscure visibility frequently, and (iii) most containers and liquid solutions lack appearance cues, therefore determining status changes relies heavily on the accurate understanding of protocols; tracking in BioLabs is a multimodal video understanding challenge that demands fine-grained comprehension of both modalities.

Table 2: **Statistics of data and annotations used for TransST and MultiAR. segmap** denotes segmentation maps in each data split. **brf_exp** and **prc_exp** denote the brief and practical experiments. **sol** denotes solutions. We use the suffix **.cls** to indicate the number of annotation categories for certain data categories and the suffix **.num** to indicate the number of annotated instances in that data category.

| | ambiguity | hours | frame | segmap | brf_exp.cls | prc_exp.cls | hoi.cls | hoi.num | obj.cls | obj.num | action.cls | action.num |
|---|---|---|---|---|---|---|---|---|---|---|---|---|
| **MultiAR** | easy | 5.1 | 17485 | 75371 | 11 | 52 | 155 | 22965 | 36 | 22965 | 20 | 22965 |
| | medium | 2.81 | 7890 | 33205 | 13 | 19 | 63 | 10651 | 13 | 10651 | 16 | 10651 |
| | hard | 1.74 | 2492 | 8561 | 9 | 8 | 57 | 5866 | 11 | 5866 | 17 | 5866 |
| | total | 9.64 | 26259 | 112937 | 13 | 79 | 245 | 37537 | 48 | 37537 | 21 | 37537 |

| | hours | frame | segmap | brf_exp.cls | brf_exp.num | prc_exp.cls | prc_exp.num | obj. cls | obj. num | sol.cls | sol. num |
|---|---|---|---|---|---|---|---|---|---|---|---|
| **TransST** | 1.05 | 41725 | 100424 | 6 | 31 | 17 | 34 | 14 | 90888 | 12 | 40443 |

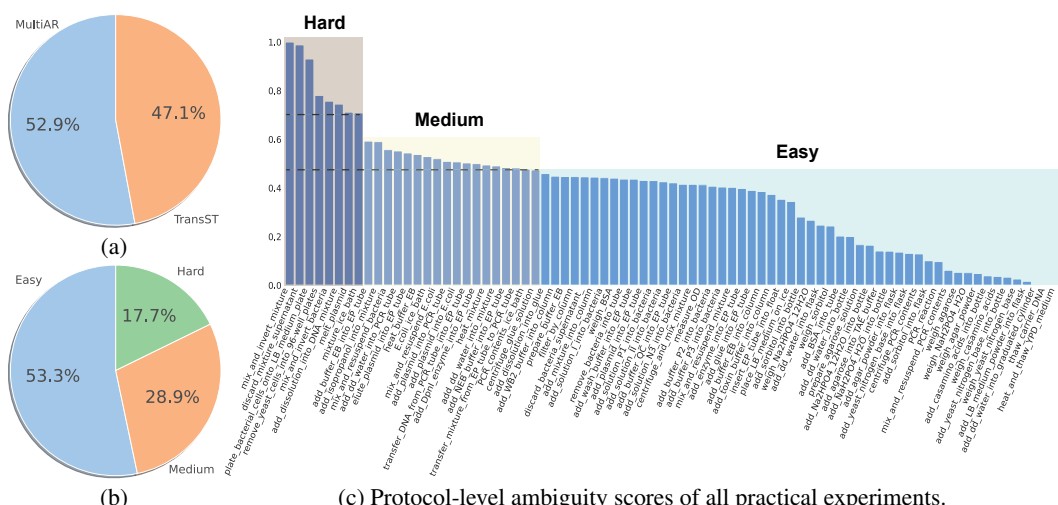

(a)

(b)

(c) Protocol-level ambiguity scores of all practical experiments.

Figure 3: (a) Despite the discrepancy between video lengths in TransST and MultiAR, we provide a comparable number of segmentation maps as ground truths in TransST for solution tracking. (b) We split the videos in MultiAR based on the ambiguity level of protocols, resulting in a 6:3:1 easy/medium/hard split. (c) One protocol is defined as *hard* when its ambiguity score surpasses 0.7, *easy* when below 0.45, and *medium* otherwise.

We evaluate the models' capability via TransST, leveraging all nearby-view videos with liquid solution labels. Each tracking problem includes the bounding box of the target object (*e.g.*, a tube) and the category label of the liquid solution inside (*e.g.*, double-distilled water). We further consider two diagnostic settings, pure visual and protocol-guided, to confirm the significance of protocols. Protocol-guided tracking leverages practical experiments as additional input to equip models with information w.r.t. invisible solution status changes. Please refer to Sec. 4.1 and Appx. B.1 for details.

**Multimodal action recognition (MultiAR)** An intelligent monitoring system in BioLabs must recognize actions and identify the corresponding protocol to track the experimental progress. However, establishing such a capability is challenging in BioLab: perceptually similar motions may have divergent semantic interpretations, and the same sub-experiment protocols across different experiments may refer to different meanings. However, current datasets have neglected the ambiguity present within fine-grained actions (Murray et al., 2012; Shao et al., 2020; Goyal et al., 2017; Kay et al., 2017; Zhu et al., 2022; Panda et al., 2017; Kanehira et al., 2018). Currently, there is no universally recognized standard for quantifying the ambiguity present in various actions. Our experiment indicates that the straightforward approach of using the similarity of human-object interactions hoi (*e.g.* Jaccard coefficient) is insufficient for adequately capturing both object ambiguity and procedural ambiguity. To address this, we propose a method for defining the ambiguity between two actions by employing the bidirectional Levenshtein distance ratio, as illustrated in Equation (1). In this equation, $P(A)$ and $P(B)$ signify the power sets of the given sets $A$ or $B$ of hoi. $ratio$ here refers to the Levenshtein distance ratio. Notably, the ambiguity (labeled as $amb$) between two practical experiments can exceed a value of 1, indicating a significant similarity between the two procedures or experiments, referred to as prc_exp. To measure the average ambiguity of each action, we then introduce a method for

Table 3: **Tracking results of all models in TransST.** We visualize the best results in bold.

| Categories | Method | PRE ↑ | NPRE ↑ | CLS ↑ | FPS ↑ | Param ↓ |
|---|---|---|---|---|---|---|
| Vision-only | TransATOM (2021a) | 27.54 | 32.20 | 29.36 | 26.0 | 7.54M |
| | YOLOv5 (2022) + StrongSORT (2022) | 47.71 | 49.49 | 42.43 | 27.9 | 86.19M |
| | YOLOv7 (2022a) + StrongSORT (2022) | 59.27 | 66.41 | 57.22 | **35.8** | **6.22M** |
| | SAM (2023) + DeAOT (2022) | 91.07 | 96.94 | 45.83 | 2.3 | 641.27M |
| Protocol-guided | YOLOv7 (2022a) + StrongSORT (2022) | 60.25 | 67.11 | 61.94 | 35.4 | **6.22M** |
| | SAM (2023) + DeAOT (2022) | **92.40** | **97.46** | **62.43** | 2.1 | 641.27M |

calculating the average ambiguity for each action, expressed mathematically as $\frac{1}{N} \sum_{amb \in N} amb_i$:

$$amb = \frac{1}{P(A)} * \sum_{x \in P(A)} \max_{y \in P(B)} \left( ratio(x, y) \right) + \frac{1}{P(B)} * \sum_{y \in P(B)} \max_{x \in P(A)} \left( ratio(y, x) \right). \quad (1)$$

As depicted in Fig. 3, there is considerable overlap among most practical experiments, leading to ambiguity when trying to distinguish them based solely on sequences of HOI. More comprehensive visual results of this phenomenon are presented in Appx. B.2. Considering the common occurrence of overlapping atomic actions, we focus on protocol-level ambiguity in MultiAR and leave perceptual-level action ambiguity as a natural intermediary challenge for models. The MultiAR benchmark is a protocol-level action recognition task with all annotated top-down view videos in 🧪 **ProBio**. We split all videos into three folds (*i.e.*, easy, medium, and hard) to evaluate protocol-level ambiguity. Over these splits, we devise four benchmarking settings: protocol-only, vision-only, vision with brief experiment guidance, and vision with detailed protocol guidance. In the protocol-only setting, we provide ground-truth HOI annotations (*i.e.*, perfect perception) to models as a performance upper bound. We add protocols of varied granularity to multimodal learning training in protocol-guided scenarios. Vision-only models are tasked to recognize protocol-level activities during testing.

## 4 Experiments

In this section, we evaluate and analyze the performance of models on tasks associated with 🧪 **ProBio**. Particularly, we provide details of the experimental setup, evaluation metrics, and result analysis for TransST and MultiAR. Fundamentally, we aim to address the following questions:

- How challenging is the fine-grained understanding of objects and actions in BioLab?
- How crucial are the protocols in tasks associated with 🧪 **ProBio**?
- What is missing in existing models when adapted to the specialized BioLab environment?

### 4.1 TransST

**Setup** As mentioned in Sec. 3.3, we consider two settings in TransST: visual tracking and protocol-guided tracking. The training, validation, and testing sets are divided in a 6:3:1 ratio, respectively, across all videos captured from nearby perspectives. In **visual tracking**, we select a number of leading-edge models to serve as our baseline comparisons, including TransATOM (Xie et al., 2020), StrongSORT with different detection backbones (Broström, 2022; Wang et al., 2022a), and Segment-and-Track-Anything (Cheng et al., 2023) based on SAM (Kirillov et al., 2023). Since SAM (Sequential Attention Model) is initially trained on general images, we adopt the strategy suggested in Chen et al. (2023) and integrate a five-layer convolutional SAM adapter. This approach is intended to adapt the SAM weights for effective application within the BioLab setting. Regarding **protocol-guided tracking**, our preliminary experiments indicate that narrowing down the category of liquid solution types to only categories mentioned in the protocols is more effective than learning-based designs (*e.g.*, fusing protocol features with tracking features). Please refer to Appx. B.1 for details.

**Evaluation metrics** Following Fan et al. (2019, 2021a), we measure the tracking quality by the precision (PRE) and normalized precision (NPRE) with an intersection-over-union (IoU) over 0.45. In addition, we evaluate the prediction of the solution status within the tracked bounding box with classification accuracy (CLS). To provide a comprehensive analysis of models, we report the memory and time overhead of all methods in transparent solution tracking.

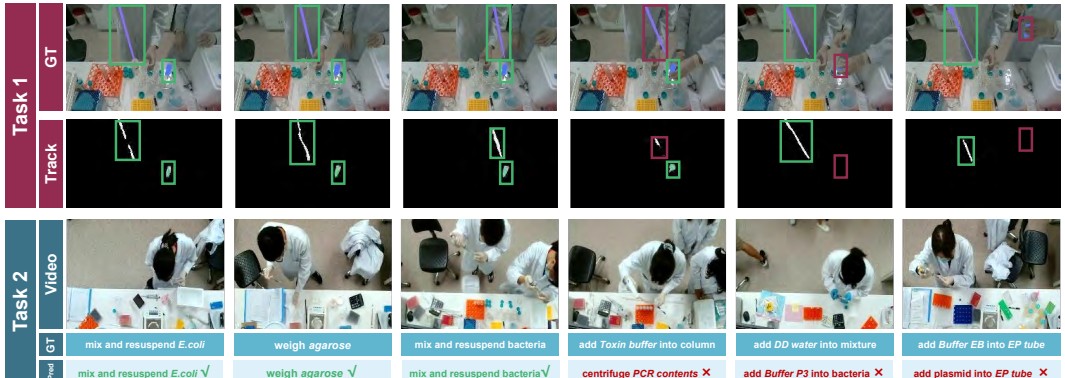

Figure 4: **Examples of success and failure cases in two tasks. Top:** Tracking results of *SAM-adapter+DeAoT* with protocol-guidance in TransRT. We visualize correct tracking predictions in **green boxes** and failure cases in **red boxes**. We observe that most failure cases could be attributed to the perceptual difficulty of transparent objects or occlusion. **Bottom:** Protocol-level action recognition results of *ActionClip+SAM* with protocol guidance in MultiAR. We visualize correct predictions in **green** and wrong ones in **red**. Of note, most incorrectly identified actions have similar HOIs.

**Results and analysis** We report transparent solution tracking results in Tab. 3 and visualize qualitative results in Fig. 4. Specifically, we summarize our major findings as follows:

- **Visual tracking in TransST is challenging.** As shown in Tab. 3, the performance of traditional (*e.g.*, StrongSORT) tracking models with only visual inputs is significantly lower than the near-perfect performance they present in common tracking scenarios (*e.g.*, driving). We have achieved higher detection efficiency while maintaining computational speed, resulting in an optimal trade-off. In TransST, solutions and containers often have transparent appearances and similar shapes that are visually difficult to distinguish. The frequent occlusion further complicates this because of containment relationship changes in biology experiments. All these facts add difficulty to the visual tracking problem in BioLab environments.

- **More robust object detectors benefit tracking in TransST.** We observe a consistent performance improvement when adopting more robust object detectors (*e.g.*, SAM). As these models are often pre-trained on large-scale object detection and segmentation datasets, we believe they are beneficial for mitigating the discrepancy between objects in daily life and specialized domains. However, limited by their memory and computation overhead, these models are still unsuitable for real-time monitoring. This urges the need for lightweight adaptations of existing pre-trained models for specialized downstream domains.

- **Understanding protocols is crucial in TransST.** As explained in Sec. 3.3, tracking the status of transparent liquid within containers is difficult as there are no direct visual features indicating the transition of liquid status. This makes label prediction for the tracked solution extremely challenging without protocol information. Our results on the classification accuracy reflect this fact, showing that event with the simplest heuristic of answer filtering, adding experimental protocols can significantly improve label prediction for all models. However, this improvement is still marginal. This implies that current models still fall short of reasoning about solution types. This promotes future research on multimodal methods for inferring visually unobservable object status changes.

## 4.2 MultiAR

**Setup** As discussed in Sec. 3.3, we evaluate performance under four distinct settings: protocol-only, vision-only, vision with brief experiment guidance, and vision with detailed protocol guidance. Similar to experiments in Sec. 4.1, we randomly split video data at each ambiguity level into train/validation/test with a 6:3:1 ratio. We evaluate the performance of BERT (Devlin et al., 2018) and SBERT (Reimers and Gurevych, 2019) in the protocol-only setting. For the vision-only scenario, we propose finetuning state-of-the-art (SOTA) video recognition models for the MultiAR task. This includes I3D (Carreira and Zisserman, 2017), SlowFast (Feichtenhofer et al., 2019), and Multiscale Vision Transformers (MViT) (Fan et al., 2021b; Li et al., 2022). For settings with detailed protocol guidance, we select strong baselines, including ActionCLIP (Wang et al., 2021), EVL (Lin et al., 2022), and Vita-CLIP (Wasim et al., 2023). We feed additional protocol information into these models

Table 4: **Experiment results of MultiAR in 🧪 `ProBio`.** We highlight the best results in bold.

| categories | method | easy | | | | mid | | | | hard | | | |
|---|---|---|---|---|---|---|---|---|---|---|---|---|---|
| | | top1 | top5 | avg. | Δ | top1 | top5 | avg. | Δ | top1 | top5 | avg. | Δ |
| Human oracle | w protocol | 98.99 | – | – | 0 | 94.34 | – | – | 0 | 93.94 | – | – | 0 |
| | w/o protocol | 64.64 | – | – | -34.35 | 56.60 | – | – | -37.74 | 51.51 | – | – | -42.43 |
| Protocol-only | BERT 2018 | 56.19 | 67.00 | 59.14 | -42.8 | 46.11 | 71.33 | 54.72 | -48.23 | 33.72 | 47.55 | 37.23 | -60.22 |
| | SBERT 2019 | 82.39 | 98.14 | 89.97 | -16.6 | 69.28 | 98.07 | 86.28 | -25.06 | 65.03 | 87.7 | 77.21 | -28.91 |
| Vision-only | I3D 2017 | 41.16 | 78.78 | 67.23 | -57.83 | 29.76 | 63.93 | 58.88 | -64.58 | 11.16 | 33.79 | 23.79 | -82.78 |
| | SlowFast 2019 | 50.03 | 89.97 | 70.07 | -48.96 | 42.16 | 67.64 | 59.72 | -52.18 | 15.00 | 42.22 | 33.08 | -78.94 |
| | MViT 2021b | 47.72 | 89.02 | 69.92 | -51.27 | 39.92 | 64.84 | 54.29 | -54.42 | 13.34 | 38.01 | 28.14 | -80.6 |
| | MViTv2 2022 | 55.28 | 91.35 | 79.92 | -43.71 | 45.25 | 69.74 | 61.24 | -49.09 | 21.37 | 46.77 | 38.94 | -72.57 |
| Protocol-guided (brief) | Vita-CLIP 2023 | 69.54 | 73.65 | 70.22 | -29.45 | 50.30 | 75.44 | 71.92 | -44.04 | 21.75 | 37.43 | 25.66 | -72.19 |
| | EVL 2022 | 72.64 | 89.74 | 80.74 | -26.35 | 55.23 | 90.74 | 81.76 | -39.11 | 36.62 | 47.75 | 39.97 | -57.32 |
| | ActionCLIP 2021 | 71.79 | 88.26 | 81.17 | -27.2 | 53.75 | 86.22 | 77.21 | -40.59 | 37.55 | 70.29 | 57.64 | -56.39 |
| | ActionCLIP 2021 + SAM 2023 | 75.27 | 95.67 | **86.05** | -23.72 | 61.77 | **92.45** | 84.24 | -32.57 | 44.54 | 76.62 | **69.22** | -49.4 |
| Protocol-guided (detailed) | Vita-CLIP 2023 | 67.25 | 74.48 | 70.57 | -31.74 | 51.76 | 79.02 | 66.49 | -42.58 | 41.25 | 67.82 | 53.33 | -52.69 |
| | EVL 2022 | 73.44 | 88.75 | 81.46 | -25.55 | 53.34 | 91.27 | 69.74 | -41 | 39.64 | 63.34 | 52.05 | -54.3 |
| | ActionCLIP 2021 | 73.93 | 93.67 | 82.23 | -25.06 | 59.42 | 84.27 | 80.01 | -34.92 | 40.7 | 71.11 | 57.67 | -53.24 |
| | ActionCLIP 2021 + SAM 2023 | **76.75** | **97.94** | 85.79 | **-22.24** | **62.79** | 91.24 | **84.40** | **-31.55** | **46.75** | **79.97** | 67.64 | **-47.19** |

during training. Considering that the text encoders utilized in these models are typically trained on text from general domains, we substitute them with SentenceBERT, which has been specifically fine-tuned in the protocol-only setting. In light of experimental findings in Sec. 4.1, we also explore the use of strong object segmentation models (*e.g*., SAM) for the multimodal understanding problem in MultiAR by adding an additional object branch into current models. We provide more model design and implementation details in Appx. B.2.

**Evaluation metrics** In all experiments, we report the recognition performance with the top-1, top-5, and mean accuracy. Additionally, we conduct human evaluations and provide the average performance of 10 experienced biology researchers with and without protocols. We report the difference (Δ) between the performance of each method and the human oracle to visualize the gap on all ambiguity levels in MultiAR.

**Results and analysis** We present model performance results of the MultiAR task in Tab. 4 and provide qualitative results in Fig. 4. In summary, we identify the following major findings:

- **Actions are visually ambiguous in MultiAR.** As shown in Tab. 4, both human and vision-only models suffer from perceptual-level ambiguity in actions. Without protocol information, we observe a 40% human performance drop in recognizing the correct protocol being executed. This indicates that humans largely depend on protocols to distinguish visually similar actions. This is also reflected by the low performance of SOTA video recognition models on the hard split with pure vision input.

- **Recognition in MultiAR demands a detailed understanding of protocols.** In contrast to other multimodal video understanding benchmarks, leveraging pre-trained language models is insufficient for MultiAR due to the specialized domain. As shown in the protocol-only setting of Tab. 4, fine-tuning a pre-trained BERT model results in low overall performance. Meanwhile, we observe a significant improvement in SBERT with better modeling of protocol contexts. This suggests potential improvements from more powerful language models that can capture fine-grained experimental contexts illustrated within protocols.

- **Contextual information is crucial for multimodal understanding in MultiAR.** As shown in Tab. 4, the stark contrast between protocol-guided models and pure vision-based models verifies the importance of contextual information in video action recognition, regardless of the protocol granularity. Although we only provide protocols during training, this suggests that it is critical for models to align visual perceptions with fine-grained protocols. Meanwhile, improving the granularity of protocol guidance generally improves model performance, especially on the hard split of MultiAR. This reveals the potential of more fine-grained multimodal interaction designs in models for improving protocol-level action recognition in MultiAR.

- **Solving protocol-level ambiguities is a bottleneck for multimodal video understanding in MultiAR.** Across the three ambiguity levels, we observe a significantly lower performance of most models in the MultiAR hard split. Intuitively, with perceptual-level ambiguity in recognizing actions, identifying protocols depends on matching typical and recognizable actions between visual

inputs and protocols. Protocol-level ambiguities aggravate this condition by adding variety in the protocols that could be matched. Nonetheless, as humans can accurately identify the protocol being executed in videos when provided with protocols, we argue that it is essential to mitigate this performance gap with better multimodal video understanding and reasoning designs.

## 5    Conclusion

**What is 🧪 `ProBio`?**    We curate 🧪 `ProBio`, the first protocol-guided dataset, which includes comprehensive hierarchical annotations in BioLab. Our dataset aims to promote protocol standardization and the development of intelligent monitoring systems to address the reproducibility crisis in biology science. We've collected 180.6h of videos within an internationally recognized molecular biology laboratory and meticulously annotated all the instruments and solutions as 213,361 segmentation maps. Additionally, we provided annotations for the conditions and enclosures of 48 transparent objects and the state of 12 solutions in 1.05h nearby-view videos. We further restructure all 9.64h videos from a top-down view into three hierarchical levels: *i.e.*, 13 `brf_exp`, 3,724 `prc_exp`, and 37,537 `hoi`. This arrangement allows for fine-grained multimodal data and categorical information insight for future research on intelligent monitoring systems.

**What can we do with 🧪 `ProBio`?**    Based on the fine-grained multimodal dataset, we devise two benchmarking tasks associated with 🧪 `ProBio`, aimed at enhancing multimodal video understanding. These two tasks are referred to as transparent solution status tracking (TransST) and multimodal action recognition (MultiAR). In TransST, we provide the transparent object's bounding box and the *<liquid category, object id>* paired labels. We further discuss the difference between pure-vision tracking and protocol-guided tracking to explore the contribution of procedure information in fundamental tasks. In MultiAR, perceptually similar motions may have divergent semantic interpretations, and the same sub-experiment protocols across different experiments may refer to different meanings. We quantify the definition of ambiguity and establish multimodal and protocol-guided action recognition to enhance the importance of fine-grained contextual information. Our experiments consistently highlight the value of such contextual information in both TransST and MultiAR tasks. Furthermore, this dataset paves the way for other tasks like video segmentation, task prediction, and procedural reasoning.

**What will 🧪 `ProBio` contribute?**    🧪 `ProBio` is the pioneering multimodal dataset captured within a molecular biology lab, aiming to enhance video comprehension by providing contextual information for visual data. The newly introduced protocol not only enhances the model's ability to process sequences over time but also incorporates the idea of "procedure." This helps our model to effectively handle action interpretations that may be ambiguous. For building a proficient monitoring system, a model skilled in understanding multimodal videos is crucial. Especially in a sterile biology lab, such a system becomes a superior tool to ensure that researchers follow the prescribed standards and procedures. The failure to reproduce experiments is frequently attributed to unintentional mistakes committed by experimenters. Within the 13 experimental categories in 🧪 `ProBio`, a notable quantity of operational actions are not executed as intended, and errors have been identified. The implementation of a monitoring system that possesses advanced video comprehension skills holds the promise of rapidly notifying experimenters of their anomalies, thereby enhancing the overall reproducibility of experiments. Consequently, this serves as a deterrent against the squandering of several months' worth of work and significant financial resources amounting to tens of thousands of dollars.

**Limitation and future work**    Currently, we considered two distinct tasks associated with 🧪 `ProBio`. However, these two tasks do not adequately reflect the myriad challenges present in today's biological laboratory environments, nor do they showcase the broad applicability and potential of our dataset. In an upcoming version of 🧪 `ProBio`, we plan to add new tasks like video-text retrieval, video segmentation, task prediction, and procedural reasoning, along with more comprehensive annotations. Regarding the model structure, we have not closely aligned the procedure information with the video feature, indicating the potential for further improvement in action recognition accuracy. Our efforts will concentrate on models, aiming to extend their capabilities to recognize and interpret actions with higher levels of detail and complexity.

**Acknowledgement** The authors would like to thank Xuan Zhang (Helixon Inc.) and Ningwan Sun (Helixon Inc.) for professional data annotation, Ms. Zhen Chen (BIGAI) for designing the figures, Tao Pu (SYSU) for assisting the experiments, and NVIDIA for their generous support of GPUs and hardware. This work is supported in part by the National Key R&D Program of China (2022ZD0114900), an NSFC fund (62376009), the Beijing Nova Program, and the National Comprehensive Experimental Base for Governance of Intelligent Society, Wuhan East Lake High-Tech Development Zone.

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

# A Data

In this section, we introduce our dataset construction process, covering both data collection and annotation. We will provide insights into our data sources, collection methods, and annotation tools. We present in detail as follows:

## A.1 Data collection

**Did you include the estimated hourly wage paid to participants and the total amount spent on participant compensation?** Yes, we did. Before the annotation and human study process, compensation was prearranged and discussed with the participating individuals. A labor fee of 100 RMB per 30 minutes will be remunerated to them, with any duration less than 30 minutes being considered as half an hour. The aggregate labor charges for all individuals involved sum up to 5,000 RMB.

### A.1.1 Biology protocol

To ensure the precision and comprehensiveness of biological protocol data, the initial step involves the retrieval of a substantial number of protocols from highly regarded journals and conferences such as Cells (MDPI, 2011), Jove (JOVE, 2006), and Protocol Exchange (NATURE, 2000) for the period spanning 2022 and prior years. The aforementioned protocols represent the forefront of experimental guidelines within the realm of biology and serve as a highly appropriate foundation for establishing a standardized protocol for biological experimentation. The microscopic realm is the setting for certain biological experiments, including brain neuroscience and genetic sequencing, which are not discernible to the unaided eye. In light of this, we have identified 12,381 experiments that are amenable to oversight via a monitoring system.

The experimental protocols procured from high-ranking academic journals are notably succinct, with most protocols offering mere guidance without practical operational steps (Ioannidis, 2005; Begley and Ellis, 2012). Hence, they are denoted as brief experiments, commonly abbreviated as `brf_exp`. To render these succinct and theoretical procedures feasible, it is imperative to deconstruct them and augment them with comprehensive instructions. For instance, we can expand the *"PCR preparation"* to [*"adding dd water into the solution," "placing the PCR in an ice bath"*, *etc.* ] by breaking it down into a series of steps. These expanded protocols are commonly referred to as practical experiments and can be denoted as `prc_exp`. Doctoral students in biology from renowned institutions, including Harvard, Peking University, and Tsinghua University, were employed to perform annotation tasks. The annotation results were thoroughly verified through multiple rounds of mutual checks to ensure accuracy and completeness. As a result, the protocols that were previously only instructive can now be executed.

An online annotation tool has been developed to streamline the annotation process for annotators across the globe and facilitate real-time multiple rounds of mutual checks. We track the information of annotators and modifiers through IDs, aiming to improve the efficiency and standardization of the annotation process. The instructions for using the annotation interface and tools are shown in Fig. A1.

### A.1.2 Monitoring video

To gather a comprehensive video collection, we have partnered with an internationally recognized biological laboratory that adheres to standard protocols Nest.Bio Labs (2023). This collaboration enables us to capture the various activities involved in conducting biology experiments. This category of laboratory adheres to an international standard that mandates uniformity in both the interior and exterior appearance and design across laboratories worldwide. Unified regulations dictate the number, color, and size of workstations, the height of the ceiling, and the dimensions of the rooms. This offers a superb opportunity to broaden the global impact and augment the applicability of our 🧪 **ProBio**.

Under the supervision of experienced researchers, we conducted the process of laboratory selection and camera setup. The selected molecular biology laboratory comprises seven primary experimental stations, a refrigeration unit, and a sterile enclosure. To ensure comprehensive coverage of all operations and instruments, we deployed ten high-resolution cameras strategically positioned from a top-down perspective to minimize occlusion. Every experimental table, refrigerator, and chamber is furnished with a specialized camera for documentation. An additional camera has been installed with

a specific focus on the frequently utilized water bath during experimental procedures, to guarantee that no procedural details are impeded or overlooked during the water bath process. Furthermore, we positioned a single RGB-D camera in proximity to the experimental table and sterile chamber to record operations with a higher level of detail and a closer perspective. Following the completion of the setup, a continuous and uninterrupted silent recording plan was implemented for the ongoing experimental operations, to minimize any potential impact on the experimenters. The raw video footage collected for this study exceeded a total of 700 hours. Subsequently, the dataset was generated via post-processing techniques and annotation procedures.

## A.2 Data annotation

Before annotation, we use the semi-automated method to remove irrelevant video clips, such as clips with no human, clips with unrelated actions, *etc*. In the semi-automated filtering process, we apply YOLOv5 (Ultralytics, 2022) and OpenPose (Cao et al., 2017) to crop key video clips with related experiment instructions and operations. We then manually remove frames depicting actions unrelated to the intended focus, such as conversing, note-taking, or texting. To ensure the efficiency of pre-processing, we carefully check each clip of our filtered videos. Finally, we obtain a total of 180.6h videos.

### A.2.1 Alignment

In the process of data collection, a total of 12,381 brief experiments were acquired, along with their respective practical experiments, following necessary adjustments and completion. We also obtained a collection of raw videos spanning 180.6h; however, no connection was established between this dataset and the aforementioned data type. To establish the correlation between the aforementioned modalities, a team of master's and doctoral students from prestigious academic institutions such as Peking University, Tsinghua University, and Peking Union Medical College Hospital were recruited to conduct alignment annotation. The task of annotation entails establishing a correspondence between the present state of videos and practical experiments (*i.e.*, `prc_exp`) through the allocation of action labels, thereby enabling the subsequent annotation of more detailed actions. An offline video action annotation tool has been developed to enhance the annotation process for annotators located in various regions. The tool, depicted in Fig. A2, enables the application of diverse labels through the use of keyboard shortcuts, thereby enhancing the efficiency of the annotation process. In the course of annotating alignments, we have ascertained that the periodic occurrence of routine operations is a common phenomenon. Consequently, we opted to engage in a collaboration with expert experimenters to carefully choose a subset of video frames from the existing footage for further detailed annotations.

### A.2.2 Fine-grained annotation

Then, we employ a team of annotators and provide a two-day professional training on all BioLab instruments, solutions, and operations. After the training, we divide the current video into multiple batches of 30-50 minutes each and deliver them iteratively to the annotation team. Before each batch delivery, we provide corresponding annotation guidelines, including the IDs of the experimental personnel, the items involved in the operation, and their respective labels. We create the dataset through real-time acceptance of online annotations. After completing 12 batches of annotations, we have annotated 213,361 segmentation maps for 10.69h and summarized two characteristics in our dataset: (i) Many operations involve the combination of multiple transparent solutions to yield a new transparent solution. In experimental settings, it is customary to employ transparent and uncolored apparatus and solutions. (ii) Similar movements represent entirely different jobs and lead to divergent purposes, which is called ambiguity.

**Solution status**    Given the two main characteristics of this dataset, while also considering the huge number of segmentation maps, we divide the dataset into two major parts. We first annotate 1.05h videos to learn more about transparent objects and solutions. Following consultation with experienced experimenters, we collect 48 object categories and 12 solution categories. Instance masks and bounding boxes are employed in video annotation to denote the positions and identities of objects. We further track the location of solutions used throughout the experiments to track the status and progress of experiments. This information is annotated by providing additional labels over container object annotations (*e.g.*, ["tube_1," "LB_solution"] for test tube with LB_solution). While exporting

annotations, we use a list of labels to represent the relations between the reagent and objects (*e.g.*, ["tube_1," "LB_solution"]).

**Hierarchical structure**   As for the second part, we focus on the ambiguity in the rest of 9.64h videos. There will be a high similarity between current practical experiments. To differentiate these ambiguous actions, we have decided to further refine them at the granularity of human-object interaction pairs in the `prc_exp`. We have divided our 🧪 **ProBio** dataset into a three-level hierarchical structure, as shown in Fig. A3. At the top level, we use brief experiment (`bf_exp`) to define the overall goal of an experiment, which is only documented in the paper and works in theory, *e.g.*, "yeast transformation" and "PCR preparation." Next, we use practical experiment (`prc_exp`) to represent practical experiments in protocols which are composed of several HOIs, *e.g.*, "measure OD" and "add YPD_medium into vector." Finally, we use HOI pairs to define atomic operations (`act`) in experiments. In total, we obtain 13 `bf_exp`, 3,724 `prc_exp`, and 37,537 `act` categories. We use a triplet for HOI annotation (*e.g.*, ["human_1," ["tube_2," "hold"]]) to represent the human subject id, interacting object, and the action verb. While exporting annotations, we translate this annotation to a list of indexes to collect the relations between humans and objects (*e.g.*, [["human_1," "object_2"], "inject"]). Finally, We instruct experimenters to conduct an additional round of verification to ensure the accuracy of labels, and the relationship of `prc_exp` and `hoi` are shown in Fig. A4.

# B   Experiment

## B.1   Transparent solution tracking (TansST)

Typically, the solution observed in BioLab exhibits characteristics of being both transparent and colorless. Since the liquid can be transferred between different containers such as beakers, petri dishes, and test tubes, the geometric shape of the liquid changes according to the shape of the container it is housed. Hence, the monitoring of the solution is an arduous and potentially unattainable undertaking. The successful execution of experiments in biology laboratories is largely dependent on the transfer and fusion of solutions, making the tracking of solutions a crucial and fundamental task in the development of a monitoring system. In our 🧪 **ProBio** dataset, we obtained pairs of containers and solutions based on the experiment's protocol and annotated them, facilitating the tracking of the solutions. During the process of using various baselines for solution tracking, we have also discovered that narrowing down the category of liquid solution types to only categories mentioned in the protocols is more effective than learning-based designs (e.g., fusing protocol features with tracking features).

### B.1.1   Implementation details

In this section, we provide details on model implementation, hyperparameters selection, and environment setup. We present the details for each selected model as follows:

**Vision-only**

- **TransATOM** Following the TransATOM (Fan et al., 2021a) benchmark, we first train the transparent solution segmentation network (Xie et al., 2020) with the TransST subset of our 🧪 **ProBio** and the easy subset of Trans10K (Fan et al., 2021a) dataset on 1 NVIDIA 3090 GPU for 40 epochs. We set the initial learning rate to 0.02, batch size to 8, and extracted visual features using ResNet18. In order to remain consistent with the original text, we also choose the ATOM (Danelljan et al., 2019) as the tracker.

- **YOLOv5 + StrongSORT** Based on StrongSORT (Broström, 2022; Wang et al., 2022a), we change different detection backbones and gain final tracking results. We first finetune the yolov5n model with the TransST subset of our 🧪 **ProBio** on 1 NVIDIA 3090 GPU for 20 epochs, we have set the initial learning rate to $1 \times 10^{-5}$, batch size to 128, and the IOU threshold as 0.45. Then, we track the detected object-solution pairs with a confidence threshold of 0.25.

- **YOLOv7 + StrongSORT** Similar to the baseline *YOLOv5 + StrongSORT*, we first finetune yolov7-tiny model with the TransST subset of our 🧪 **ProBio** on 1 NVIDIA 3090 GPU for 20 epochs, we have set the initial learning rate to $1 \times 10^{-5}$, batch size to 128, and the IOU threshold as 0.45. Then, we track the detected object-solution pairs with a confidence threshold of 0.25.

- **SAM + DeAOT** Inspired by Chen et al. (2023), we train a SAM-adapter based on vit_h pre-trained weights and AdamW optimizer with the TransST subset of our 🧪 **ProBio** dataset. We have set the learning rate to $2 \times 10^{-4}$, batch size to 2. The adapter consists of two MLPs and an activate function GELU (Hendrycks and Gimpel, 2016) within two MLPs (Liu et al., 2023). We further passed the output of the adapter through a classification network, which has five *Conv2d* layers with input patch sizes of 24. We set the patch_size as 16, window_size as 14, input image resolution as 1024×1024, and train on 4 NVIDIA A100 GPUs for 20 epochs. For models with large parameter sizes like this, training adapters have shown good performance on our 🧪 **ProBio** dataset. Then, we track the detected object-solution pairs with DeAOT (Yang and Yang, 2022), choosing the model R50-DeAOT-L.

**Protocol-guided**

- **YOLOv7 + StrongSORT** Similiar to the vision-only method, we first select object-solution pairs that have occurred based on the protocol of this experiment, including `prf_exp` and `prc_exp`, and compile them into a list. Then, we finetune the yolov7-tiny model with the filtered list on 1 NVIDIA 3090 GPU for 15 epochs, we have set the initial learning rate to $1 \times 10^{-5}$, batch size to 128, and the IOU threshold as 0.45. Then, we track the detected object-solution pairs with a confidence threshold of 0.25.

- **SAM + DeAOT** Using the same approach as protocol-guided baseline *YOLOv7 + StrongSORT*, we first filter the desired object-solution pairs through a protocol and compile them into a list. Afterward, we perform model finetuning and subsequent tracking as baseline *SAM + DeAOT*.

### B.2 Multimodal action recognition (MultiAR)

As evidenced in Appx. A.2.2, motions that are perceptually similar may possess distinct semantic interpretations, and practical experiments conducted across varying protocols may pertain to dissimilar meanings. To demonstrate the protocol-level ambiguity between two protocols in an intuitive manner, we perform a calculation of the overlap of all downstream HOI annotations. Based on the computed ambiguity metric, the complete dataset has been categorized into three distinct levels of complexity: easy, medium, and hard. Given that each level encompasses distinct practical experiments `prc_exp`, we conducted separate experiments at each level and subsequently derived conclusions. Subsequently, each of them will be explicated individually.

#### B.2.1 Ambiguity

With the increased granularity of action refinement, the inherent ambiguity of actions becomes apparent. However, current datasets have neglected the ambiguity present within fine-grained actions (Murray et al., 2012; Shao et al., 2020; Goyal et al., 2017; Kay et al., 2017; Zhu et al., 2022; Panda et al., 2017; Kanehira et al., 2018). Furthermore, there is currently no widely accepted metric for measuring ambiguity in actions. We find that the simplicity of using the similarity of human-object interactions `hoi` (*e.g.*, Jaccard coefficient) to describe both the object ambiguity and procedure ambiguity is inadequate. Therefore, we define ambiguity between two actions with the bidirectional Levenshtein distance ratio, as shown in Equation (1). In Equation (1), P(A) and P(B) represent the power set of the given A or B set of `hoi`, while ratio denotes the Levenshtein distance ratio. The ambiguity (*i.e.*, $amb$) between two practical experiments can exceed 1, which represents a high similarity between the two `prc_exp` (shown in Fig. A5). Afterward, to measure the average ambiguity of each action, we define it by taking the average value (*i.e.*, $\frac{1}{N} \sum_{amb \in N} amb_i$).

$$amb = \frac{1}{P(A)} * \sum_{x \in P(A)} \max_{y \in P(B)} (ratio(x, y)) + \frac{1}{P(B)} * \sum_{y \in P(B)} \max_{x \in P(A)} (ratio(y, x)) \qquad \text{(A1)}$$

#### B.2.2 Model Structure

To enhance the proficiency of the model, it is imperative to employ the technique of variable manipulation to isolate the specific components that necessitate refinement. Initially, a comparison is made between the conversion of human-object interactions into descriptive text and pure vision. It is concluded that the visual modality presents a greater potential for enhancement. Subsequently, the model is enhanced through the incorporation of an alignment module and an object-centric mask

module, resulting in a notable enhancement of the multimodal model's performance. Ultimately, we substitute the concise instructions with hands-on experiments that furnish extensive insights for more intricate guidance. Fig. A6 depicts the particular operations, whereby spatial information about objects is incorporated via graph neural network (GNN) (Scarselli et al., 2008), and practical experimental information is incorporated via SentenceBERT (Reimers and Gurevych, 2019). The calculation of similarity is performed consistently, and subsequently, the ultimate prediction outcome is generated.

### B.2.3   Implementation details

In this section, we provide details on model implementation, hyperparameters selection, and environment setup. We present the details for each selected model as follows:

**human study**   To assess the viability of the two proposed benchmarks and establish the maximum attainable experimental performance, a human study was conducted with the participation of ten master's students hailing from UC Berkeley, Peking University, and Tsinghua University. The study was bifurcated into two parts: *with protocol* and *without protocol*. The study involved the extraction of data from video recordings at varying levels of difficulty, namely easy, medium, and hard. The amount of data extracted was equivalent to 0.05 times the total of each level, and a list of 79 practical experiments was provided for the participants to choose from. The experimental data about the section labeled as *without protocol* had already been prepared. For the *with protocol* part, additional information about the brief experiment to which the video belonged was provided to the participants to provide direction. All participants in the experiment were remunerated according to the criteria mentioned in Appx. A.1.

**Protocol-only**   First, we process the detection results of human-object interaction in the video into textual form as input for subsequent steps. We then use protocol-guided techniques to predict the actions in the target video. This method helps reduce the influence of detection errors in the video and achieve the highest performance achievable at the current stage.

- **BERT** We use the pre-trained BERT model and implementation provided by Hugging Face (Devlin, 2018). We use the Adam optimizer Kingma and Ba (2014) and apply cross-entropy loss. We set the initial learning rate to 0.02, dropout as 0.5, batch size to 8, and train with our descriptive text on 1 NVIDIA 3090 GPU for 20 epochs.

- **SBERT** Similar to BERT, we use the pre-trained SentenceBERT model and implementation provided by Hugging Face (Chiusano, 2019). Based on the current descriptive text, we connect the `hoi` using prompts to create a practical experiment with a sequence of operations. For example, *"First, we open the tube. Second, we take the pipette,* etc." The generated sentences are then used as training inputs for the model. We use the Adam optimizer Kingma and Ba (2014) and apply cosine similarity loss. We set the initial learning rate to $2 \times 10^{-5}$, batch size to 8, and train with our descriptive text on 1 NVIDIA 3090 GPU for 20 epochs.

**Vision-only**

- **I3D** Follow (Carreira and Zisserman, 2017), ResNet50 is selected as the backbone and the frames and sampling rate are set to 8. The input video undergoes a resizing process to achieve dimensions of $224 \times 224$. The Adam optimizer Kingma and Ba (2014) is employed with a weight decay of $1 \times 10^{-4}$ and a uniform batch size of 64. The present model exhibits uniform settings across three distinct categories and undergoes training through the utilization of a single NVIDIA A100 GPU, throughout 100 epochs.

- **SlowFast** Follow (Feichtenhofer et al., 2019), we also choose ResNet50 as the backbone and both the frames and sampling rate are set to 8. The input video undergoes a resizing process to achieve dimensions of $224 \times 224$. The Adam optimizer Kingma and Ba (2014) is employed with a weight decay of $1 \times 10^{-4}$ and a uniform batch size of 64. The present model exhibits uniform settings across three distinct categories and undergoes training through the utilization of a single NVIDIA A100 GPU, throughout 100 epochs.

- **MVIT** Follow (Fan et al., 2021b), we choose MViT as the backbone and set the frames as 16, and the sampling rate as 4. The input video undergoes a resizing process to achieve dimensions of $224 \times 224$. The AdamW optimizer Loshchilov and Hutter (2019) is employed with a weight decay of $5 \times 10^{-2}$ and a uniform batch size of 16. We apply soft cross entropy as the loss function.

The present model exhibits uniform settings across three distinct categories and undergoes training through the utilization of a single NVIDIA A100 GPU, throughout 100 epochs.

- **MVITv2** Follow (Li et al., 2022), we choose MViT as the backbone and set the frames as 16, and the sampling rate as 4. The input video undergoes a resizing process to achieve dimensions of $224 \times 224$. The AdamW optimizer Loshchilov and Hutter (2019) is employed with a weight decay of $5 \times 10^{-2}$ and a uniform batch size of 4. We apply soft cross entropy as the loss function. The present model exhibits uniform settings across three distinct categories and undergoes training through the utilization of a single NVIDIA A100 GPU, throughout 100 epochs.

**Protocol-guided (brief)**

- **Vita-CLIP** Follow (Wasim et al., 2023), we finetune the pretrained CLIP model with our 🧪 `ProBio` dataset on 4 NVIDIA A100 GPUs for 50 epochs. The Adam optimizer Kingma and Ba (2014) is employed with a weight decay of $5 \times 10^{-2}$ and a uniform batch size of 64. We set the initial learning rate to $4 \times 10^{-4}$, and the frames and sampling rate as 8.

- **EVL** Follow (Lin et al., 2022), we finetune the pretrained CLIP model with our 🧪 `ProBio` dataset on 4 NVIDIA A100 GPUs for 50 epochs. The Adam optimizer Kingma and Ba (2014) is employed with a weight decay of $5 \times 10^{-2}$ and a uniform batch size of 64. We set the initial learning rate to $4 \times 10^{-4}$, the frames as 32, and the sampling rate as 8.

- **ActionCLIP** Follow (Lin et al., 2022), we finetune the pretrained ViT-B model with our 🧪 `ProBio` dataset on 1 NVIDIA 3090 GPU for 40 epochs. The AdamW optimizer Loshchilov and Hutter (2019) is employed with a weight decay of $2 \times 10^{-1}$ and a uniform batch size of 4. We set the initial learning rate to $5 \times 10^{-6}$, the frames as 32, and the sampling rate as 8.

- **ActionCLIP + SAM** We have the same vision branch and similarity calculation module as baseline *ActionCLIP*. Furthermore, we encode the object information with the graph neural network (GNN). The encoder contains two parts: temporal and spatial, each composed of MLPs with different layers, and ultimately outputs object features of 256 dimensions. After that, it is concatenated with the image feature and inputted into the subsequent loss calculation and backpropagation module.

**Protocol-guided (detailed)**  The input caption of the model was modified by replacing its text modality with a practical experiment (`prc_exp`) connected by prompts. This modified input was then passed to the encoder as a text sequence. Subsequently, the text encoder in the model was substituted with SentenceBERT. The training input and associated particulars about this segment of the model have been expounded upon in great detail within this passage (refer to Appx. B.2.3). The following is a list solely comprised of hyperparameters:

- **Vita-CLIP** The Adam optimizer Kingma and Ba (2014) is employed with a weight decay of $5 \times 10^{-2}$ and a uniform batch size of 64. We set the initial learning rate to $4 \times 10^{-4}$, and the frames and sampling rate as 8.

- **EVL** The Adam optimizer Kingma and Ba (2014) is employed with a weight decay of $5 \times 10^{-2}$ and a uniform batch size of 64. We set the initial learning rate to $4 \times 10^{-4}$, the frames as 32, and the sampling rate as 8.

- **ActionCLIP** The AdamW optimizer Loshchilov and Hutter (2019) is employed with a weight decay of $2 \times 10^{-1}$ and a uniform batch size of 4. We set the initial learning rate to $5 \times 10^{-6}$, the frames as 32, and the sampling rate as 8.

- **ActionCLIP + SAM** The AdamW optimizer Loshchilov and Hutter (2019) is employed with a weight decay of $2 \times 10^{-1}$ and a uniform batch size of 4. We set the initial learning rate to $5 \times 10^{-6}$, the frames as 32, and the sampling rate as 8.

## C  Ethical review

**Did you describe any potential participant risks, with links to Institutional Review Board (IRB) approvals, if applicable?**  Yes, we did. We captured the daily experimental operations of the researchers through ten cameras fixed on the ceiling, filming in a 24-hour uninterrupted silent mode. We obtained consent from all personnel involved in the experiment and applied blur to the recorded faces to ensure the confidentiality of personal information. During the data recording period, no specific actions were required from the participants, and we submitted a complete set of materials

to the Institutional Review Board (IRB), including the list of subjects, experimental details, duration, and all relevant materials.

### C.1 Responsibility & data license

We bear all responsibility in case of violation of rights and our dataset is under the license of CC BY-NC-SA (Attribution-NonCommercial-ShareAlike).

## D Future work

Currently, regarding the two benchmarks proposed in this article, we have demonstrated the effectiveness of detailed protocol-guided for complex video understanding through experiments. Our plans for model structure, data annotation, and task enhancement are outlined. Furthermore, expanding the applicability of our dataset is a priority for us. To this end, we aim to develop a monitoring system using our current multimodal dataset. This system is designed to reduce the occurrence of experimental errors by experimenters, improve the repeatability and correctness of experiments, curtail expenses, and augment efficacy.

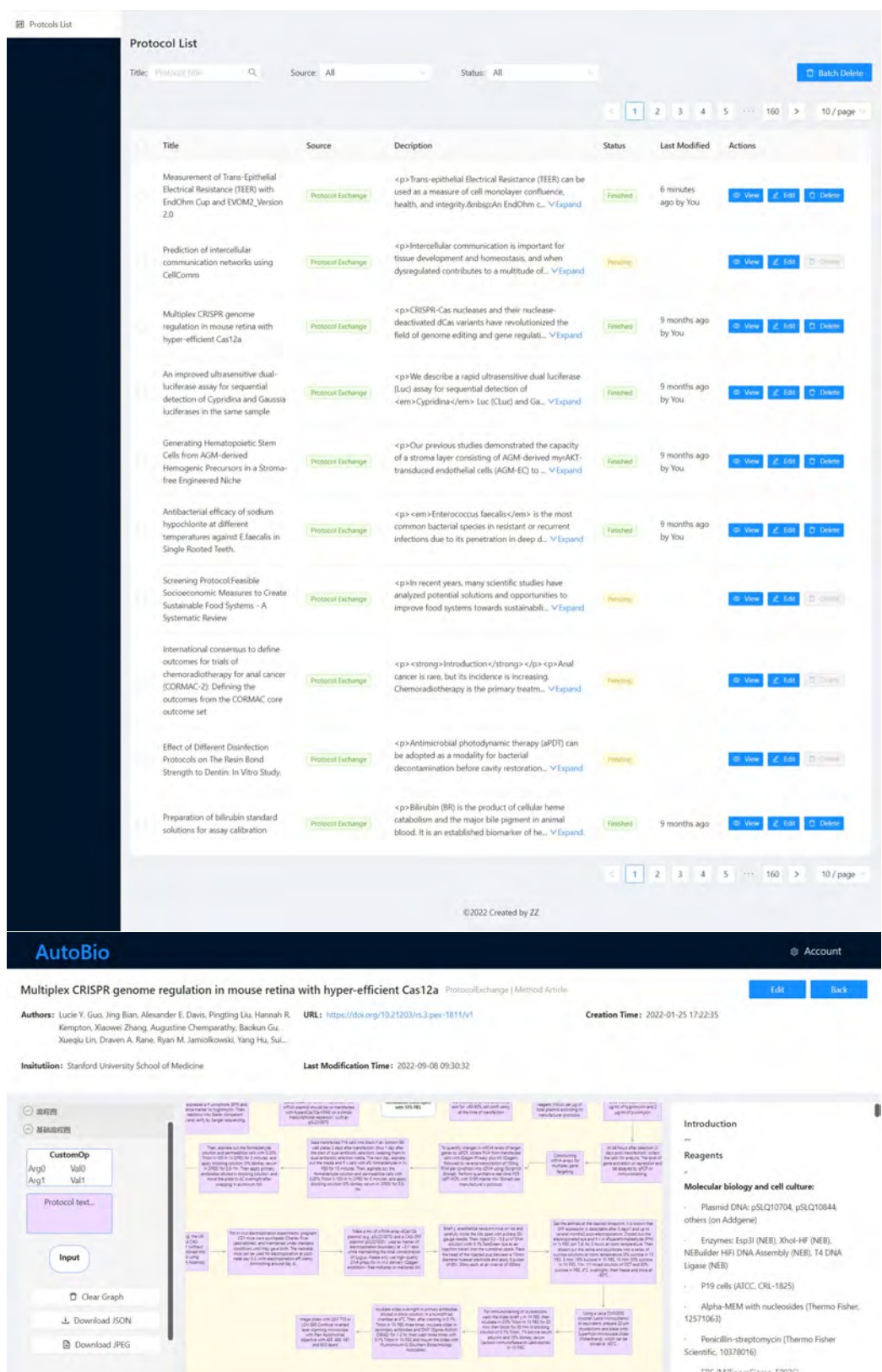

Figure A1: (a) The home page. The logged user needs to choose the target protocols and whether to view or edit. (b) The annotation page. Protocol details are shown on the top of the page, and annotators need to complete the annotation process through multiple clicks, dragging, and input operations.

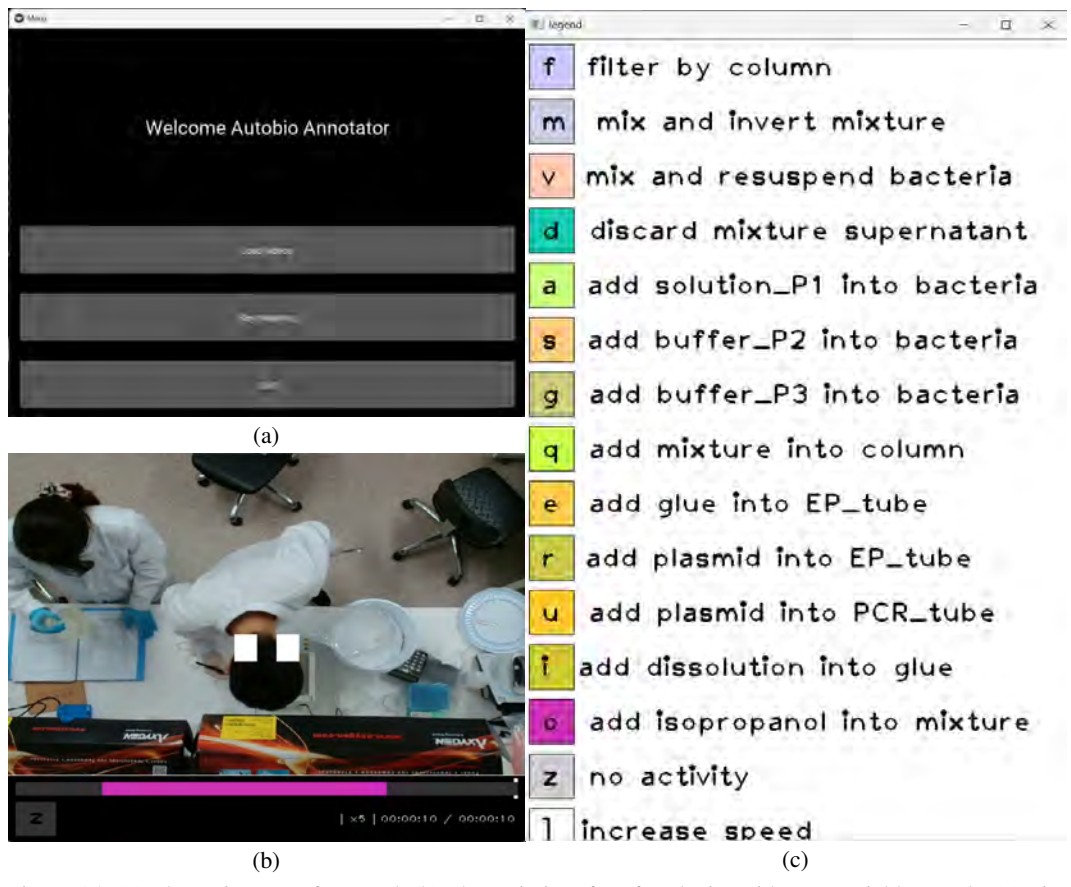

Figure A2: (a) The main page of our tool. (b) The main interface for playing videos at variable speeds. (c) List of `prc_exp` of the chosen `brf_exp`.

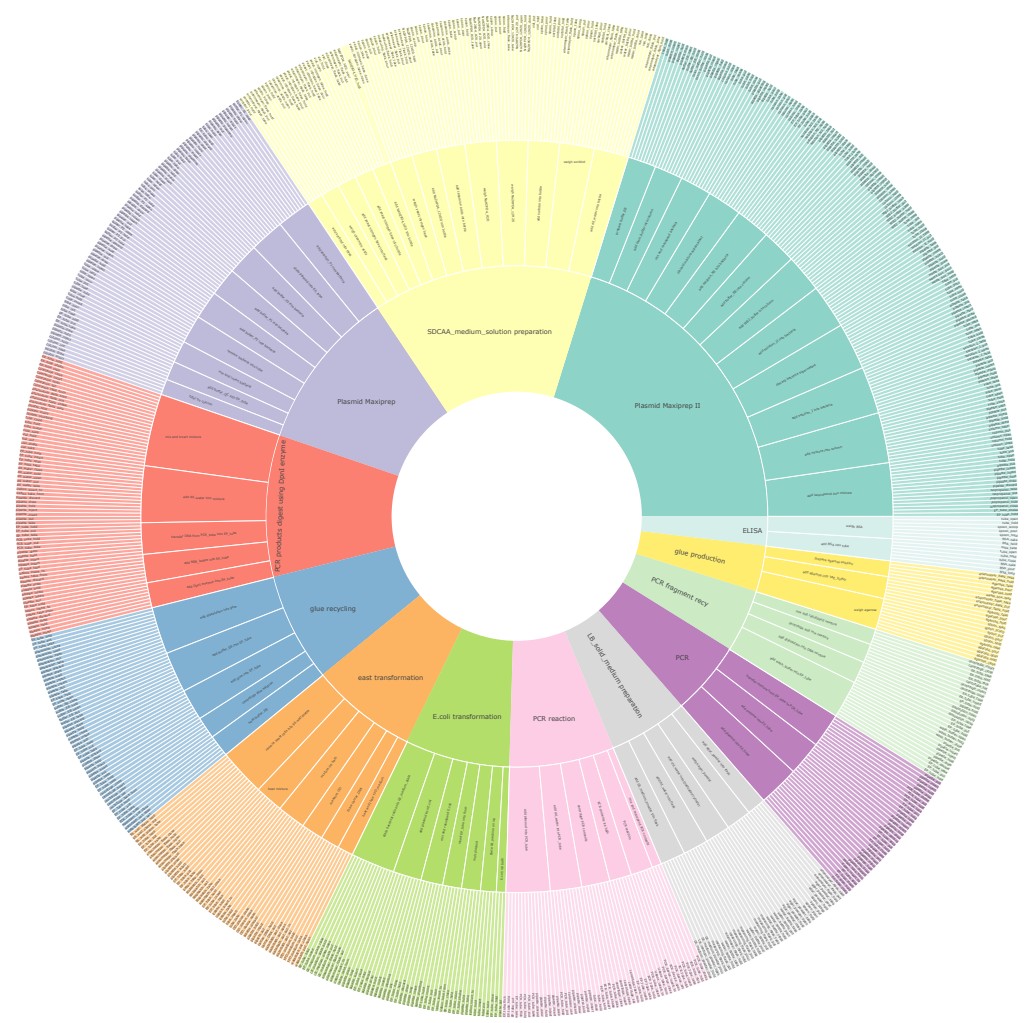

Figure A3: The three-level hierarchical structure.

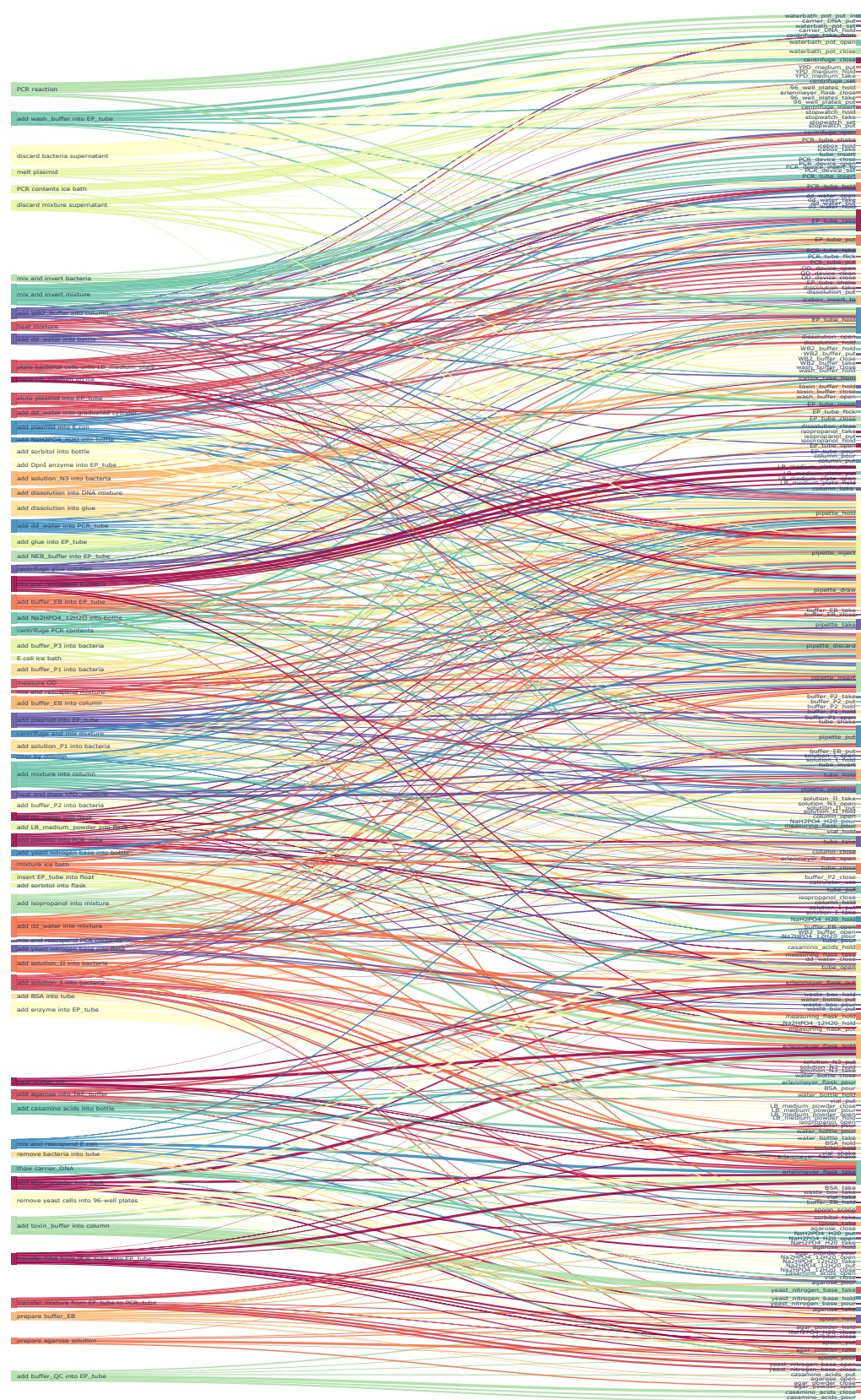

Figure A4: Relationship between `prc_exp` and `hoi`.

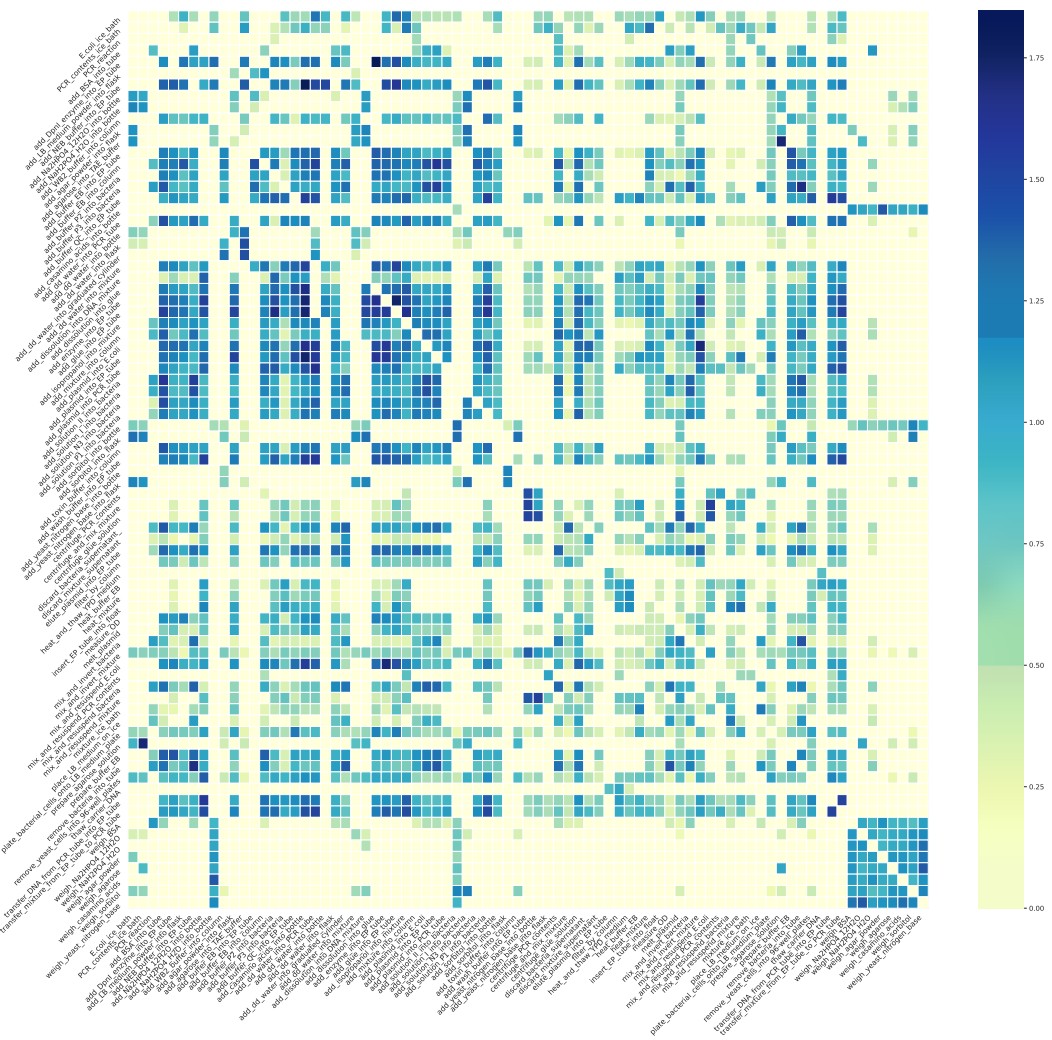

Figure A5: Visualization of the ambiguity between any two actions.

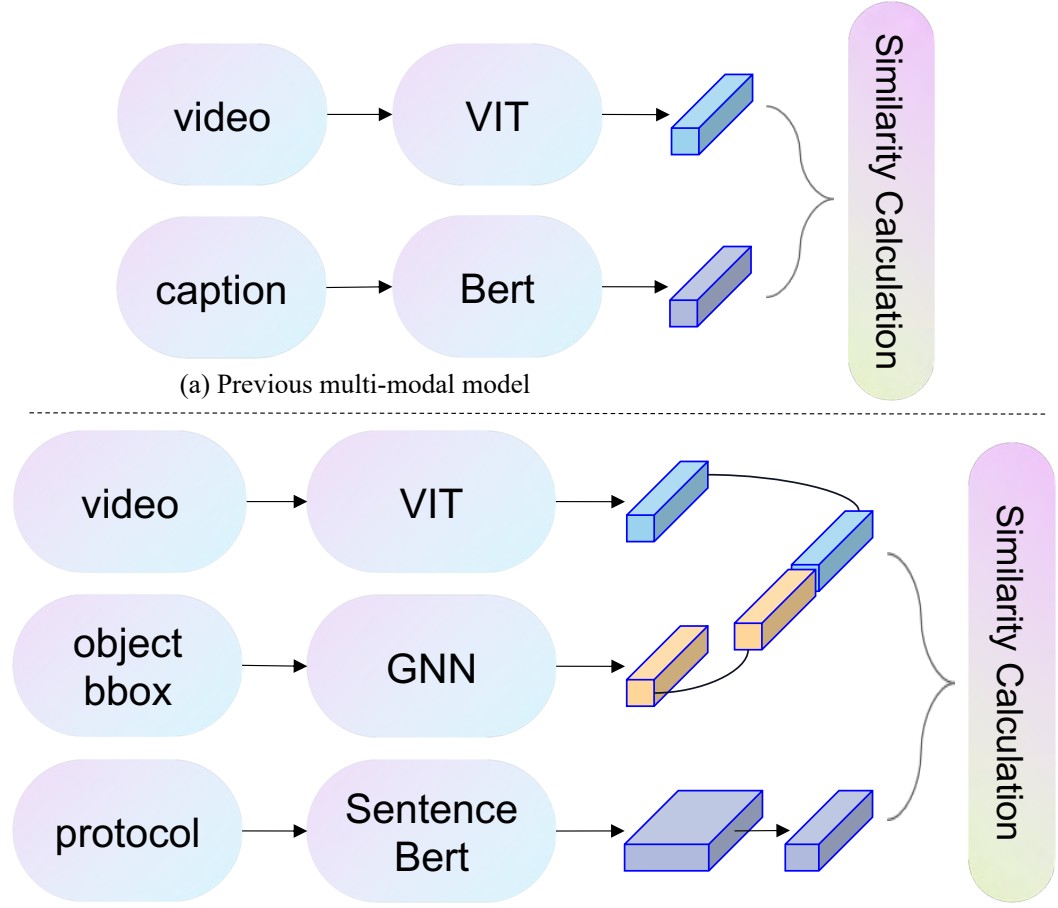

(a) Previous multi-modal model

(b) Ours multi-modal model

Figure A6: Structure of our action recognition module

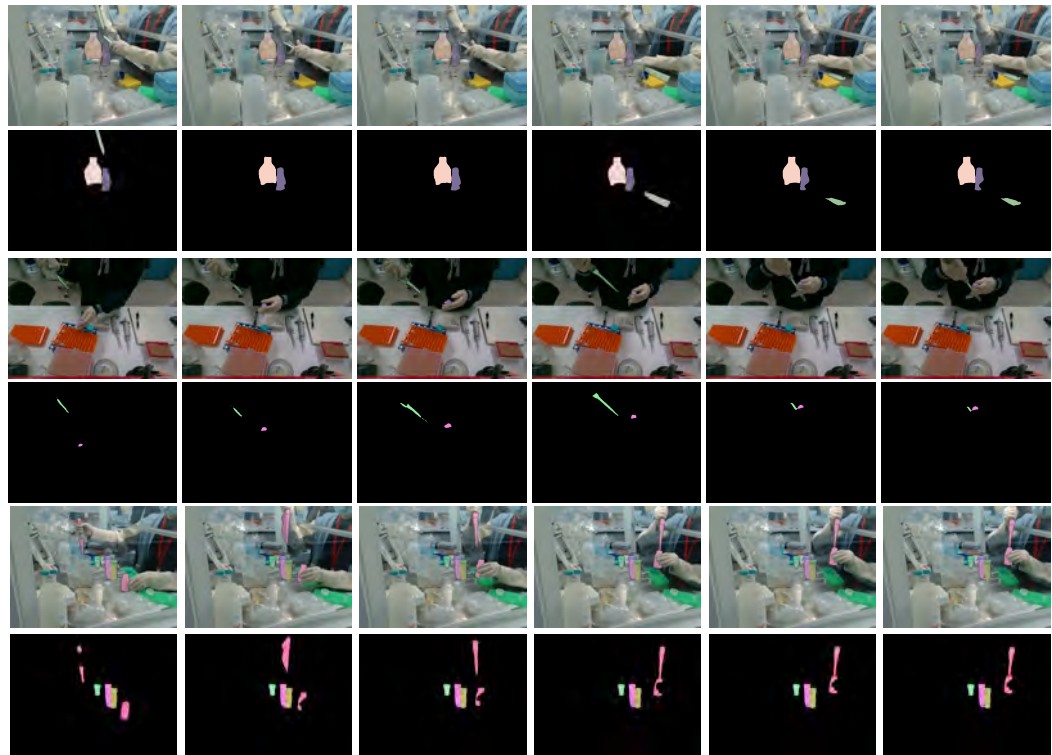

Figure A7: Visualization of the TransST results.

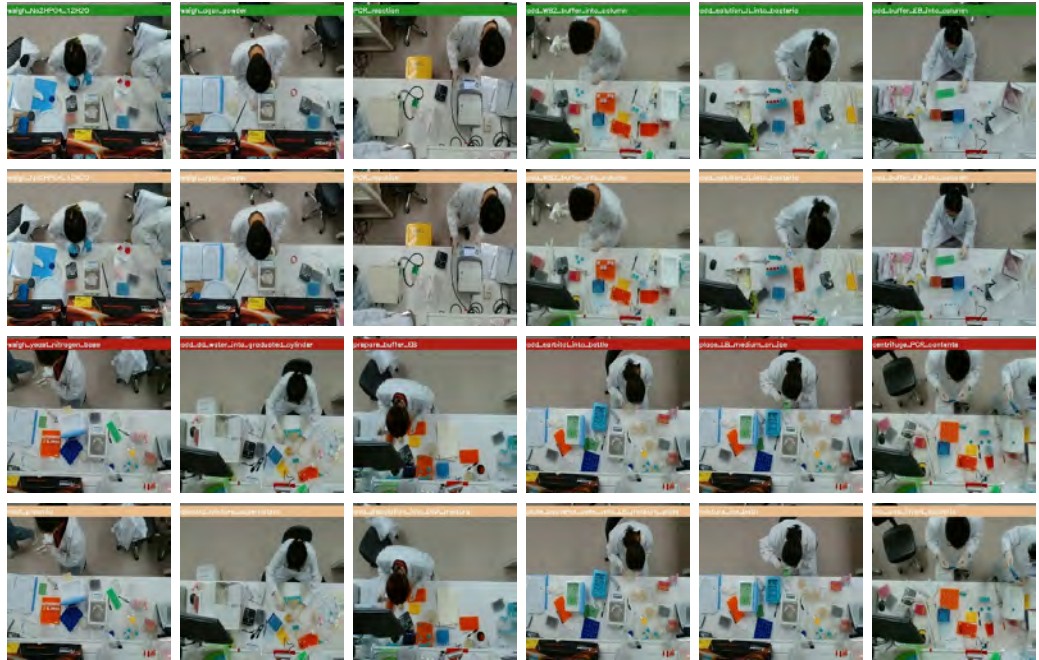

Figure A8: Visualization of the MultiAR results.

