# OpenReview forum: "ProBio: A Protocol-guided Multimodal Dataset for Molecular Biology Lab"
_NeurIPS.cc/2023/Track/Datasets_and_Benchmarks — NeurIPS 2023 Datasets and Benchmarks Poster_

### Official Review · Reviewer_CFqS · 2023-07-16
**Review for ProBio: A Protocol-guided Multimodal Dataset for Molecular Biology Lab**

**Rating:** 7
**Confidence:** 2
**Correctness:** The manuscript looks correct to me

**Strengths:**

- The paper is well written and the motivation is clearly stated. The lack of robust model for lab protocol tracking can be attributed to the lack of available dataset
- The comprehensiveness for the breakdown of the labels, including different biology experiments, the activities involved to conduct them, and the human-object interactions, are clearly stated and backed up by reasonable intuitions for the usage in biology research
- The benchmark design is comprehensive. They employ most of the state-of-the-art methods for video understanding as well as suitable text encoders for the protocol learning

**Additional Feedback:**

N/A

**Clarity:**

- The ambiguity score was never defined but the authors start using it to describe their experiments

**Documentation:**

The documentation looks sufficient to me

**Limitations:**

- The high-level intuition of using video to help biology research's reproducibility is understandable. However, how does such dataset enable that is a bit unclear. Specifically, what's the benefit when a model can recognize the actions conducted in a biology experiment? If it's simply for reproducibility then why not just record the actions so that the next researcher knows what to follow. These intuitions are not clearly stated in introduction.
- The transparent solution tracking and action recognition seems to be describing the same thing, the fine-grained activities on conducting the biology experiment. One reader might suspect the transparent solution tracking task is to learn the status changes of the solutions, e.g. volume changes, but not the activities exerted on the solutions.

**Opportunities For Improvement:**

- The intuitions to incorporate the written protocol in protocol-guided tracking lacks explanations. If protocols are provided in real life, why do we need a model to recognize the specific actions to conduct those protocols?
- Also, it seems like adding protocols doesn't benefit all metrics across the board. What's the reasoning FPS metric has stronger performance on vision-only method?



**Relation To Prior Work:**

It's unclear why the authors compare their multimodal dataset for biology research to something like sports, in table 1. The nature of actions in the biology lab should be more fine-grained, where they should compare to other fine-grained activity recognition datase

**Summary And Contributions:**

This paper presents a multimodal dataset for molecular biology research, mainly focusing on video understanding for the lab protocols conducted. The overall presentation is clear, where the authors make arguments for a carefully curated dataset for the successful replication of the molecular biology research. The data is collected in the lab through real life research activities and then annotated by both automatic algorithms and human labor. The tasks they focus on mainly has two categories, transparent solution tracking and multimodal action recognition. They employ strong vision models, specifically for object detection and segmentations, for a comprehensive benchmarking.

---

> ### Author Response · Authors · 2023-08-25
> **Response to Reviewer CFqS (1/3)**
>
> We thank the reviewer for acknowledging our experimental design and also the effectiveness of our dataset. Please refer to the general response for a summary of all changes we have made in this revision. We further clarify the reviewer's concerns as follows:
>
> **Q1: FPS metric has stronger performance on the vision-only method**
>
> FPS (frames per second) refers to the number of frames that the model can process within a single second. As shown in Table 3, we acknowledge that adding textual modalities to a vision-only setup increases computational overhead, which consequently lowers the FPS. However, metrics also show that after adding protocol-guided elements, the overall computational speed of the model does not significantly decrease. This allows us to achieve higher detection efficiency while maintaining computational speed, resulting in an optimal trade-off. We have clarified this in our revised paper.
>
>
> **Q2: Why do we need protocols and models to recognize the action, rather than the recorded video?**
>
> Thank you for your question. We believe our system offers several benefits over traditional methods (e.g., video-recording), as outlined below:
> 1. **Due to different degrees of expertise, operators may implement given protocol differently.** In such situations, a real-time monitoring system is crucial for supervising ongoing actions, utilizing accumulated data from seasoned operators to increase overall success and reproducibility rates.
> 2. **When tasks are conducted manually, human error is always present.** There is no assurance that following the steps shown will result in success. Cooking videos, for instance, may aim to teach everyone how to prepare gourmet meals, but results can vary greatly, particularly with laboratory procedures, which are frequently more intricate and nuanced. Even if you follow a lab video step-by-step, errors in actions or omissions can lead to unsuccessful reproducibility.
> 3. **Uncontrollable cost.** Even assuming that protocols are perfectly detailed and people can correctly follow videos, we would still need to produce specific instructional videos for each experiment, setting, and operator. Obtaining such data is prohibitively expensive compared to developing a monitoring system that can continuously incorporate new data and enhance itself.
>
>
> In summary, we believe that our monitoring system offers higher efficiency and lower costs than traditional methods. Thank you again for raising this important question, which has encouraged us to think more deeply about the real-world implications of our work, thereby improving the quality of our paper.

---

> > ### Author Response · Authors · 2023-08-25
> > **Response to Reviewer CFqS (2/3)**
> >
> > **Q3: How will the dataset help to improve the reproducibility in BioLab**
> >
> > For improving the reproducibility in BioLab, we first collect the ProBio dataset and devise two challenging benchmarks. Specifically, we aim to improve the reproducibility issue with the following contributions:
> > 1. **Protocol-guided monitoring system.** The failure to reproduce experiments is frequently attributed to unintentional mistakes committed by experimenters. Within the 13 experimental categories in ProBio dataset, a notable quantity of operational actions were not executed as intended and errors have been identified. The implementation of a monitoring system that possesses advanced video comprehension skills holds the promise of rapidly notifying experimenters of their anomalies, thereby enhancing the overall success rate of reproducibility. Furthermore, in the context of a sterile biological lab, the monitoring system serves as the best tool for ensuring that experimenters' operations and procedures meet established standards. Consequently, this serves as a deterrent against the squandering of several months' worth of work and significant financial resources amounting to tens of thousands of dollars.
> > 2. **Video-understanding models that address ambiguous actions.** Creating an effective monitoring system requires a model with advanced skills in interpreting multimodal videos. However, understanding actions and tracking solutions in a biological lab environment present unique challenges. In these settings, solutions are often transparent and lack a defined shape, human-object interactions (hoi pairs) come in numerous combinations, and similar-looking movements can indicate entirely different jobs and lead to divergent purposes. These issues of transparent solutions and ambiguous actions are prevalent in virtually every phase of biological experiments. To improve the model's performance in complex BioLab, we introduce two specific yet challenging benchmarks: transparent solution tracking and ambiguous action recognition.
> > 3. **Regularized biological protocol design.** The introduced protocol serves to improve the temporal processing capabilities of our video comprehension model. It also introduces the concept of 'procedure',  which allows our model to more effectively interpret actions that might be ambiguous. While it can be both costly and time-consuming to standardize brief protocols gathered from inconsistent online sources, this standardization is crucial. It minimizes unnecessary experimentation and failures, thereby improving both the reproducibility and accuracy of experiments.
> >
> >
> > **Q4: Difference between two benchmarks**
> >
> > Thank you for your inquiry into the details of our benchmark tasks; your questions will undoubtedly help us improve the quality of our work. First, the tasks were indeed designed to focus on fine-grained activities involved in conducting biology experiments. However, each task emphasizes different aspects in our benchmark design.
> >
> > - **Task1 Transparent Solution Tracking**, centers on instance segmentation of tiny objects as well as tracking entirely transparent and shapeless solutions. This task poses significant challenges within the field of computer vision, necessitating an in-depth comprehension of visual features and their related textual inputs. Nevertheless, the tracking of solution state changes is crucial in biological experiments.
> > - **Task2 Multimodal Action Recognition**, focuses on ambiguous actions. The aim is to distinguish visually similar actions by understanding data across multiple modalities. The addition of knowledge on the spatial positioning of tiny objects, along with protocol guidance, has the potential to enhance the rate of recognition accuracy to approximately 70%. This statistic underscores the challenging nature of this task and elucidates why there is currently a dearth of comparable datasets available.
> >
> > As for concerns about changes in object states, including volume changes, temperature changes, and time variance, which are indeed factors that directly affect the success rate of biological experiments. At the moment, we are constrained by certain factors. For example, the arrangement of our equipment should not interfere with normal operations, and the equipment cannot be worn by the operators. We can only collect data from the bird-view, which makes it challenging to monitor numbers on the equipment, as they may only occupy 1-2 pixels in the image.  Going forward, we plan to collaborate with worldwide laboratories and set up cameras over the experimental tables to capture hand movements, aiming to acquire more detailed experimental data and further enhance the quality of our dataset.

---

> > > ### Author Response · Authors · 2023-08-25
> > > **Response to Reviewer CFqS (3/3)**
> > >
> > > **Q5: More details about ambiguity**
> > > We have added a detailed explanation about ambiguity and provided the formulation. This content has been added to Section **3.3 benchmark** in the main text and is highlighted in blue. More visualization results are updated in the project page (i.e. https://probio-dataset.github.io/dataset.html).
> > >
> > > >Similar movements represent entirely different jobs and lead to divergent purposes, which is named ambiguity. This ambiguity isn't limited to just biological experiments; it extends to our everyday experiences as well. Consider a situation where someone is waving their hand. In such cases, it's unclear whether the person is intending to greet, bid farewell, or signal a negative response.
> > >
> > > >With the increased granularity of action refinement, the inherent ambiguity of actions becomes apparent. However, current datasets have neglected the ambiguity present within fine-grained actions. Furthermore, there is currently no widely accepted metric for measuring ambiguity in actions. We find that the simplicity of using the similarity of human-object interactions <tt>hoi</tt> (i.e. Jaccard coefficient) to describe both the object ambiguity and procedure ambiguity is inadequate. Therefore, we define ambiguity between two actions with the bidirectional Levenshtein distance ratio, as shown in the following equation. P(A) and P(B) represent the power set of the given A or B set of <tt>hoi</tt>, while ratio denotes the Levenshtein distance ratio. The ambiguity (i.e. $amb$) between two practical experiments can exceed 1, which represents a high similarity between the two <tt>prc\_exp</tt>. Afterward, in order to measure the average ambiguity of each action, we define it by taking the average value (i.e. $\frac{1}{N} {\sum \limits_{amb\in N} amb_i}$).
> > >
> > > > $amb = \frac{1}{P(A)}*\sum \limits_{x\in P(A)}{\underset{y\in P(B)}{\max}(ratio(x, y))}$
> > >
> > > > $ + \frac{1}{P(B)}*\sum \limits_{y\in P(B)}{\underset{x\in P(A)}{\max}(ratio(y, x))}$
> > >
> > >
> > >
> > > **Q6: Why we compare with sports dataset**
> > >
> > > Thank you for your question. Here are the reasons we compare with sports-related and kitchen-related datasets, such as FineDiving[1] and FineGYM[2]:
> > > - **Related annotation of fine-grained actions.** These datasets are among the most fine-grained datasets currently available for video action understanding. For example, FineGYM proposes a three-hierarchical structure of *events, sets, and elements (i.e., Balance Beam -> Flight handspring -> Flic-flac with 0.5 twist to handstand)*. FineDiving breaks down diving actions into detailed components like *mid-air rotation angles and poses*, and also provides procedure-aware information.
> > > - **Related task of fine-grained actions.** Although the subject matter of these papers may appear to differ significantly from our area of focus in BioLab operations, which is inherently more intricate and complex, there are similarities in both the implementation process and overall methodology. COIN[3], FineGYM, and YouCook2[4] all offer evaluation tasks related to action recognition. Furthermore, there is a current lack of datasets with more fine-grained action details, which is also a goal we are striving to achieve. Therefore, we find the comparison to be informative and relevant.
> > >
> > > > [1] Shao D, Zhao Y, Dai B, et al. Finegym: A hierarchical video dataset for fine-grained action understanding[C]//Proceedings of the IEEE/CVF conference on computer vision and pattern recognition. 2020: 2616-2625.
> > > > [2] Xu J, Rao Y, Yu X, et al. Finediving: A fine-grained dataset for procedure-aware action quality assessment[C]//Proceedings of the IEEE/CVF Conference on Computer Vision and Pattern Recognition. 2022: 2949-2958.
> > > > [3] Tang Y, Ding D, Rao Y, et al. Coin: A large-scale dataset for comprehensive instructional video analysis[C]//Proceedings of the IEEE/CVF Conference on Computer Vision and Pattern Recognition. 2019: 1207-1216.
> > > > [4] Zhou L, Xu C, Corso J. Towards automatic learning of procedures from web instructional videos[C]//Proceedings of the AAAI Conference on Artificial Intelligence. 2018, 32(1).

---

> > > > ### Comment · Reviewer_CFqS · 2023-08-30
> > > >
> > > > I have read the responses and appreciate the authors for addressing my questions.

---

### Official Review · Reviewer_bbUW · 2023-07-21
**Review of ProBio**

**Rating:** 6
**Confidence:** 4

**Strengths:**

The strenghths of this paper are three-fold:

- Significance: ProBio tackles an important real-world problem of improving reproducibility in biology experiments. The dataset enables developing intelligent monitoring systems to address this.
- Relevance: Video understanding in specialized domains like labs is an underexplored area. ProBio provides a brand-new and comprehensive multi-view video dataset with extensive annotations following a rigorous protocol augmentation and data collection process.
- Social impact: This paper has the potential to improve integrity in biological research through intelligent monitoring systems developed using this dataset. It remains possible to enable discoveries and innovations in the field.

In summary, the key strengths of this paper are the significance of improving reproducibility for research findings in the molecular biology lab, the relevance of dataset for specialized video understanding, the complexity of data collection and annotations, and the beneficial social impacts this work could enable. The collaboration with biology experts also helps strengthen its value for the field.

**Additional Feedback:**

I am intrigued by the potential for this benchmark to facilitate the reproducibility of research findings in the biology lab. While on the surface it is a specialized computer vision dataset, the alignment of fine-grained annotations to standardized protocols provides a bridge to addressing the critical issue of reproducibility, both in biology experiments and AI assistance. A more satisfactory explanation connecting the benchmark to replicability in the domain could further highlight the broader impacts. Specifically, discussing how models trained on this data could be integrated into intelligent monitoring systems to track proper protocol adherence would help substantiate the claims around reproducibility. Exploring this connection between the dataset and the real-world implications in more depth could lead me to assign a higher score when recommending this paper by better supporting the significance of the work.

**Clarity:**

Some typos:

- The caption in Table 1: "denotethe" should be "denote the";


**Correctness:**


Based on the information provided in the paper submission:

- The critical claims around introducing a new multimodal biology lab dataset and associated benchmarks appear supported. The data collection, annotation, and benchmark construction are documented.
    - For the dataset: The protocol curation, video recording, filtering, and annotation processes seem rigorous and sound. The hierarchical annotations aligned to protocols are a novel contribution.
    - For the benchmarks: The tasks of solution tracking and action recognition are logically motivated by the domain challenges. The evaluation metrics, splits, and experiments appear appropriately designed and executed.

- The analysis of model performance shortcomings is evidence-based on the benchmark results.

- The limitations around long-term hosting plans and ethical usage guidelines point to areas for improvement. But the core contribution claims seem sound.

In summary, the key claims around the proposed dataset and benchmarks appear to be supported by the information presented. The dataset construction and benchmark design choices are reasonable. The claims seem to accurately reflect the contributions, though some additional dataset details could further strengthen the work.

**Documentation:**

Based on the information provided in the paper:

- Data Collection and Organization: The data collection process is described in detail, including protocol curation, video recording setup, and configuration, filtering and processing of raw video, and annotation methodology.
The hierarchical data organization with fine-grained HOI labels mapped to protocols is clearly explained.
Overall, the dataset construction seems sufficiently documented.

- Availability and Maintenance: The paper provides a URL to access the dataset. However, details are not provided on long-term hosting plans and maintenance. This information would help support ongoing access.

- Ethical and Responsible Use: The collaboration with biology experts helps ensure protocols are conducted responsibly. Usage guidelines could help promote the ethical use of the data. But specific policies are not mentioned. Evaluation metrics, splits, and results are documented for the benchmarks to support reproducibility.

In summary, data collection and organization are sufficiently detailed, but more information on long-term availability and ethical usage guidelines would further strengthen the dataset contribution. The benchmarks seem sufficiently detailed.

**Ethics:**

No.

**Limitations:**

See above in **Opportunities For Improvement**.

**Opportunities For Improvement:**

The real-world impact relies on future work to develop intelligent monitoring systems. The connection between the dataset and improving reproducibility could be more concretely demonstrated.

**Relation To Prior Work:**

It is clearly discussed in section 2.

**Summary And Contributions:**

- This paper introduces ProBio, a new multimodal dataset for molecular biology labs with $180$ hours of multi-view video and fine-grained annotations aligned to protocols.
- This paper provides benchmarks for transparent solution tracking and multimodal action recognition. The analysis shows current models underperform on these tasks, indicating opportunities for future work to improve video understanding for biology experiments.

---

> ### Author Response · Authors · 2023-08-25
> **Response to Reviewer bbUW**
>
> We are grateful for your clear summary of our ProBio dataset and your recognition of the beneficial social impacts this work could enable. Please refer to the general response for a summary of all changes we have made in this revision. Here we provide further discussions on the reviewer's concerns:
>
> **Q1: How will the dataset help to improve the reproducibility in BioLab**
>
> For improving the reproducibility in BioLab, we first collect the ProBio dataset and devise two challenging benchmarks. Specifically, we aim to improve the reproducibility issue with the following contributions:
> 1. **Protocol-guided monitoring system.** The failure to reproduce experiments is frequently attributed to unintentional mistakes committed by experimenters. Within the 13 experimental categories in ProBio dataset, a notable quantity of operational actions were not executed as intended and errors have been identified. The implementation of a monitoring system that possesses advanced video comprehension skills holds the promise of rapidly notifying experimenters of their anomalies, thereby enhancing the overall success rate of reproducibility. Furthermore, in the context of a sterile biological lab, the monitoring system serves as the best tool for ensuring that experimenters' operations and procedures meet established standards. Consequently, this serves as a deterrent against the squandering of several months' worth of work and significant financial resources amounting to tens of thousands of dollars.
> 2. **Video-understanding models that address ambiguous actions.** Creating an effective monitoring system requires a model with advanced skills in interpreting multimodal videos. However, understanding actions and tracking solutions in a biological lab environment present unique challenges. In these settings, solutions are often transparent and lack a defined shape, human-object interactions (hoi pairs) come in numerous combinations, and similar-looking movements can indicate entirely different jobs and lead to divergent purposes. These issues of transparent solutions and ambiguous actions are prevalent in virtually every phase of biological experiments. To improve the model's performance in complex BioLab, we introduce two specific yet challenging benchmarks: transparent solution tracking and ambiguous action recognition.
> 3. **Regularized biological protocol design.** The introduced protocol serves to improve the temporal processing capabilities of our video comprehension model. It also introduces the concept of 'procedure',  which allows our model to more effectively interpret actions that might be ambiguous. While it can be both costly and time-consuming to standardize brief protocols gathered from inconsistent online sources, this standardization is crucial. It minimizes unnecessary experimentation and failures, thereby improving both the reproducibility and accuracy of experiments.
>
>
> **Q2: Additional information on long-term hosting plans and dataset downloads**
>
> Our long-term hosting plan for ProBio dataset is divided into four key phases: Initial Setup, Monitoring and Maintenance, Yearly Review, and Long-Term Maintenance.
> - We have already finished the initial phase.
> - Monitoring and Maintenance involve real-time tracking of storage, unauthorized access, and regular data backups.
> - A yearly review assesses performance, costs, and data integrity.
> - In the long-term, the focus shifts to scalability, and regular audits.
>
> This comprehensive plan aims to address all aspects from security and ensure the dataset is effectively managed over its lifecycle. We have taken careful steps to de-identify and organize a subset of ProBio, which includes annotations on segment maps, solution relations, and human-object interactions. For instructions on how to obtain the dataset, please consult the 'Download' section on our project website: https://probio-dataset.github.io/. More visualization results and splits are updated in the dataset page (i.e. https://probio-dataset.github.io/dataset.html).

---

### Official Review · Reviewer_7ke8 · 2023-07-21
**Novel dataset for a complex task**

**Rating:** 9
**Confidence:** 4
**Correctness:** The labeling seems to be acceptable.
**Clarity:** The writeup is relatively clear.

**Strengths:**

A rather extensive annotated dataset collected in a controlled lab environment to assist with action recognition and activity tracking.

**Additional Feedback:**

Some of the references based on which claims are made are pretty old, going as far as 1987.

**Documentation:**

Documentation seems to be OK.

**Ethics:**

I do not think the paper acknowledges any ethical concerns.

**Limitations:**

This is a pretty comprehensive dataset, its use needs to be explored to fully comprehend its capabilities, limitations, and opportunities for improvement.

**Opportunities For Improvement:**

This is a pretty comprehensive dataset, its use needs to be explored to fully comprehend its capabilities, limitations, and opportunities for improvement.

**Relation To Prior Work:**

Prior work is clearly discussed.

**Summary And Contributions:**

The presented dataset is a protocol-guided with hierarchical annotations in biolab facility collection of videos. The authors also propose two benchmarking tasks and provide their experimental analysis. It appears to be first of its kind dataset collected in a bioresearch lab environment for the purpose of characterizing the activities of the lab, specifically 13 common biolab experiments.

---

> ### Author Response · Authors · 2023-08-25
> **Response to Reviewer 7ke8**
>
> Thank you for your constructive and encouraging feedback on our manuscript. We are delighted to hear that you recognize the novelty and value of our ProBio dataset, which, to the best of our knowledge, is indeed the first of its kind collected in a biological lab environment. We have made three primary contributions:
> 1. We introduce, ProBio, the first protocol-guided dataset with dense hierarchical annotations in BioLab to facilitate the standardization of protocols and the development of intelligent monitoring systems for reducing the reproducibility crisis.
> 2. We propose two challenging benchmarking tasks to measure models' capability in leveraging both visual observations and language protocols for fine-grained multimodal video understanding, especially for ambiguous actions and environment states.
> 3. We provide an extensive experimental analysis of the proposed tasks to highlight the limitations of existing multimodal video understanding models and point out future research directions.
>
> The benchmarking tasks and experimental analyses are created to give both practical and brief protocols, and we are glad that these contents resonated with you. Thank you once again for your valuable feedback. Your acknowledgement of our focus on 13 popular BioLab experiments encourages us to continue our research. We are also actively seeking collaborations to increase the amount and diversity of our dataset, with the ultimate aim of making even more significant contributions to the field.

---

### Official Review · Reviewer_7iDD · 2023-07-21
**A multi-modal dataset for molecular biology lab workflows**

**Rating:** 6
**Confidence:** 3
**Correctness:** The work presentation is sound.
**Clarity:** The clarity is good.

**Strengths:**

- Very relevant topic: reproducibility in experimental sciences is a big concern and the work provided by authors motivates to create datasets and machine learning methods to alleviate it.
- Strong data collection and annotation effort
- Clear paper structure, easy to follow

**Additional Feedback:**

minor typos:

Contextual information are crucial --> is crucial

**Documentation:**

Datasheet for the dataset is provided in the supplementary. More extensive documentation does not seem to exist. Webpage link https://probio-dataset.github.io/ is incorrectly reported in the main paper (https://probio.github.io)

**Ethics:**

Authors provide sufficient report on ethical procedures necessary to conduct dataset collection.

**Limitations:**

The number of recorder experiments is rather low (13), it might be though enough to attract community to the dataset and benchmarks introduced to add more in succession.

**Opportunities For Improvement:**

- Authors mention the potential of their work to improve reproducibility in experimental lab setting in the introduction a lot. It would be good to have some discussion on how authors see the outcomes of their work with relation to that goal.
- only two different benchmarking tasks may be not sufficient to get community attracted to experiment with the dataset.

**Relation To Prior Work:**

Prior work is properly reflected.

**Summary And Contributions:**

In the presented work, authors curate a multi-modal dataset for molecular biology lab workflows - ProBio - recorded in a laboratory adhering to international standards, containing video and experiment protocol data.

To obtain experimental protocols, authors crawl publicaly available protocols from top-tier journals and conferences, which contain high level instructions authors refer to as brief experiments (brf_exp). They use an online annotation tool to get further detailed experimental instructions from researchers (practical experiments, prc_exp).

Recording in total over 700 hours of video through 24 hours monitoring with multiple cameras arranged to capture the lab scenario, authors make use of further filtering employing OpenPose and YOLOv5 to get to 180.6 hours containing selection of 13 brief experiments.

Authors further generate Human Object Annotations (HOIs) for the collected videos, using a representative subset consisting of a 9.64 hours top-down view videos and 1.05 hours nearby view videos. This results in three level hierarchical annotation, containing a list of 48 objects and 21 action verbs in HOIs based on the significance confirmed by 147 seasoned biology researchers.

To demonstrate value of the collected data, authors present two benchmarking tasks.  Transparent solution tracking (TransST) deals with classifying and tracking liquid solutions involved in experiments. Multimodal action recognition (MultiAR) tests for the ability to recognize relevant actions executed during lab procedures.

Using both benchmarking tasks, authors obtain results that hint on importance of multi modal protocol guidance, as protocol-guided multi modal models are observed to outperform purely vision guided or those models that rely on language only.

Authors mention that dataset is a first step towards solving reproducibility issues in experimental molecular biology and conclude that on the model side, there is still a lot of work required to bring the performance necessary for extracting essential experimental procedures from the available data.

Contributions are as following:

* Collection and curation of ProBio, a protocol-guided dataset with dense hierarchical annotations recorded in BioLab, with aim of reducing the reproducibility crisis.
* Setting up two benchmarking tasks to measure capability of currently available models in leveraging both visual observations and language protocols for handling information available in such multimodal data
* An experimental analysis of the proposed tasks with regard of the limitations of existing multimodal video understanding models

---

> ### Author Response · Authors · 2023-08-25
> **Response to Reviewer 7iDD**
>
> We appreciate your acknowledgment about the quality and contributions of our dataset. Please refer to the general response for a summary of all changes we have made in this revision. We further clarify the reviewer's concerns as follows:
>
> **Q1: The relation between our dataset and reproducibility improvement**
>
> For improving the reproducibility in BioLab, we first collect the ProBio dataset and devise two challenging benchmarks. Specifically, we aim to improve the reproducibility issue with the following contributions:
> 1. **Protocol-guided monitoring system.** The failure to reproduce experiments is frequently attributed to unintentional mistakes committed by experimenters. Within the 13 experimental categories in ProBio dataset, a notable quantity of operational actions were not executed as intended and errors have been identified. The implementation of a monitoring system that possesses advanced video comprehension skills holds the promise of rapidly notifying experimenters of their anomalies, thereby enhancing the overall success rate of reproducibility. Furthermore, in the context of a sterile biological lab, the monitoring system serves as the best tool for ensuring that experimenters' operations and procedures meet established standards. Consequently, this serves as a deterrent against the squandering of several months' worth of work and significant financial resources amounting to tens of thousands of dollars.
> 2. **Video-understanding models that address ambiguous actions.** Creating an effective monitoring system requires a model with advanced skills in interpreting multimodal videos. However, understanding actions and tracking solutions in a biological lab environment present unique challenges. In these settings, solutions are often transparent and lack a defined shape, human-object interactions (hoi pairs) come in numerous combinations, and similar-looking movements can indicate entirely different jobs and lead to divergent purposes. These issues of transparent solutions and ambiguous actions are prevalent in virtually every phase of biological experiments. To improve the model's performance in complex BioLab, we introduce two specific yet challenging benchmarks: transparent solution tracking and ambiguous action recognition.
> 3. **Regularized biological protocol design.** The introduced protocol serves to improve the temporal processing capabilities of our video comprehension model. It also introduces the concept of 'procedure',  which allows our model to more effectively interpret actions that might be ambiguous. While it can be both costly and time-consuming to standardize brief protocols gathered from inconsistent online sources, this standardization is crucial. It minimizes unnecessary experimentation and failures, thereby improving both the reproducibility and accuracy of experiments.
>
>
> **Q2: More tasks to increase attractiveness**
>
> During the benchmark development phase, we tried various tasks including tiny object detection, containment instance segmentation, human re-identification, upper-body pose estimation, and video-text retrieval. The task settings and corresponding fine-grained annotations will be made available in a future iteration of our **ProBio** dataset.
>
> **Q3: The number of recorder experiments is rather low(13）**
>
> The current version of our ProBio dataset includes 13 practical experimental protocols. These were carefully distilled from more than 700 hours of video footage, gathered over nearly a year. The dataset covers a variety of key biology experiments focused on yeast transformation, among others. To ensure the data's accuracy, we devoted four months and a budget of 9,549 USD to carefully review and update the annotations.
> Meanwhile, we're actively reaching out to labs around the globe to keep expanding the dataset. Our goal is to encourage more scientists to contribute, so we can all benefit from standardized and reliable experimental methods.

---

### Official Review · Reviewer_P3VE · 2023-07-26
**ProBio**

**Rating:** 8
**Confidence:** 3
**Clarity:** Yes

**Strengths:**

The depth of the annotation of the dataset here is extremely impressive, and the addition of scientific tasks to the other categories of activities studied with these kinds of problems (sports, cooking, etc) is a nice and interesting change. The authors show a broad variety of approaches tested.

**Additional Feedback:**

Very cool project!

**Correctness:**

As someone with >10 years active biology lab work, the fact that the segmentation was done by people with only a 2 day "bootcamp" training to me seems potentially problematic, as expertise may be needed to assess many of these steps accurately.  I am also concerned that 200K segmentation maps and about 80K labels (possibly - it's not clear if this was compensated, it kind of seems like it can't have been) were created in only 25 hours of work (per the authors discussion of the compensation - 5000 RMB at 200 RMB per hour) - I worry some or much of the data may not be of very high quality. Access to the data will be critical here.

**Documentation:**

Data and code were not actually yet provided, at least not in a way that allows anonymous reviewer access - the website in question only has a link to a name-and-email collecting reviewer form, and the GitHub still shows pushing the code as a TODO. So not yet possible to fully assess that, though the answers to some questions (such as hosting plan and license) are provided.

**Limitations:**

If the idea is indeed improve scientific reproducibility, is the idea that future biology labs will all be outfitted with monitoring at this level of detail? That seems somewhat dystopian to me; if the authors are suggesting it, the harms should also be raised.

**Opportunities For Improvement:**

I think the point about ambiguity and how much can be expected for very similar tasks is a good one - it is under-explained though, it would be nice to be explained in more details as well as some conclusions drawn.

**Relation To Prior Work:**

Yes

**Summary And Contributions:**

The authors here describe a large dataset of video tracking of biology experiments, annotated at multiple hierarchical levels. Based on these annotations, they set several benchmarks for multiple classes of tasks based on architectures popular in these kinds of tasks.

---

> ### Author Response · Authors · 2023-08-25
> **Response to Reviewer P3VE (1/2)**
>
> We gratefully thank you for acknowledging the effectiveness of our proposed dataset and the comprehensive experiments in our benchmark. Please refer to the general response for a summary of all changes we have made in this revision. We further make the following clarifications to address the reviewer's concerns:
>
> **Q1: More details about ambiguity**
>
> We have added a detailed explanation about ambiguity and provided the formulation. This content has been added to Section **3.3 benchmark** in the main text and is highlighted in blue. More visualization results are updated in the project page (i.e. https://probio-dataset.github.io/dataset.html).
>
> >Similar movements represent entirely different jobs and lead to divergent purposes, which is named ambiguity. This ambiguity isn't limited to just biological experiments; it extends to our everyday experiences as well. Consider a situation where someone is waving their hand. In such cases, it's unclear whether the person is intending to greet, bid farewell, or signal a negative response.
>
> >With the increased granularity of action refinement, the inherent ambiguity of actions becomes apparent. However, current datasets have neglected the ambiguity present within fine-grained actions. Furthermore, there is currently no widely accepted metric for measuring ambiguity in actions. We find that the simplicity of using the similarity of human-object interactions hoi (i.e. Jaccard coefficient) to describe both the object ambiguity and procedure ambiguity is inadequate. Therefore, we define ambiguity between two actions with the bidirectional Levenshtein distance ratio, as shown in the following equation. P(A) and P(B) represent the power set of the given A or B set of hoi, while ratio denotes the Levenshtein distance ratio. The ambiguity (i.e. $amb$) between two practical experiments can exceed 1, which represents a high similarity between the two prc\_exp. Afterward, in order to measure the average ambiguity of each action, we define it by taking the average value (i.e. $\frac{1}{N} {\sum \limits_{amb\in N} amb_i}$).
>
> > $ amb = \frac{1}{P(A)}*\sum \limits_{x\in P(A)}{\underset{y\in P(B)}{\max}(ratio(x, y))}$
>
> > $ + \frac{1}{P(B)}*\sum \limits_{y\in P(B)}{\underset{x\in P(A)}{\max}(ratio(y, x))}$
>
> **Q2: The harm of monitoring systems in the future**
>
> The objective of the monitoring system was to ensure the accuracy and reliability of experimental outcomes, **rather than to exert control over the experimenters themselves**. In designing and implementing our system, our primary aim was to *detect potential errors or inconsistencies within the experimental process*. To underscore this point, it's essential to note that our system is specifically calibrated to focus solely on activities directly associated with biological experiments. Actions or decisions made by experimenters that fall outside the purview of these experiments, especially those unrelated or tangential, will be deliberately overlooked by our system. This demonstrates our commitment to supporting a free and independent research environment while merely providing a safeguard against potential experimental discrepancies.

---

> > ### Author Response · Authors · 2023-08-25
> > **Response to Reviewer P3VE (2/2)**
> >
> > **Q3: The details of the time and cost involved in annotation**
> >
> > In the supplementary materials, we briefly mentioned the commissions involved in the annotation process. However, we only detailed the process supervised by experts and the associated costs. Here, we supplement more details on the annotation and cost:
> >
> > 1. A cost of ~$700 was spent on hiring professional biologists for annotation and human studies. During this approximately 25 hours, the participants completed tasks that included normalizing protocol actions, filtering video clips, and manually identifying various tasks in the test subset. As experts in the project, they also participated in our discussions over several months.
> > 2. In addition, the 37,537 HOI annotations and 213,361 object segmentation maps mentioned in the article were annotated by another group of annotators we collaborated with. We conducted a two-day training session for these annotators to familiarize them with the names and physical characteristics of the items, operations, and liquids required for annotation. We designed an online delivery platform for data annotation, where annotators can complete multiple category annotations.
> >
> > Furthermore, in order to guarantee the precision of the annotations:
> > 1. We initially divided the entire dataset into 16 distinct batches;
> > 2. We performed meticulous evaluations on an annotation platform that was accessible to both the biology specialists and the annotation team, scrutinizing each occurrence individually. The number of frames in each batch varied between 1,000 and 13,000;
> > 3. We promptly rectified annotation issues pertaining to categories and relationships and returned inaccurate results for re-annotation.
> > 4. To assure the final quality of the data, we conducted an average of 2-3 iterations every batch. The total amount paid for the annotators and professionals was **9,549 USD**.
> >
> > **Q4: The access of our dataset and code**
> >
> > Due to the sensitivity of our dataset, it may be necessary to obtain the dataset through a form submission in order to mitigate potential misuse of the data. We have taken careful steps to de-identify and organize a subset of ProBio, which includes annotations on segment maps, solution relations, and human-object interactions. For instructions on how to obtain the dataset, please refer to the 'Download' section on our project website: https://probio-dataset.github.io/. We will organize and open-source all the code and models as soon as possible.

---

> > > ### Comment · Reviewer_P3VE · 2023-08-30
> > >
> > > I thank the authors for their responses to my questions.

---

### Author Response · Authors · 2023-06-20
**URL**

Our website is https://probio-dataset.github.io

---

### Author Response · Authors · 2023-08-25
**The general response to reviewers**

We thank all reviewers for their insightful comments and acknowledgment of our contributions. We follow comments and suggestions from all reviewers and revise our manuscript accordingly as follows:

1. Additional explanation regarding the relationship between our dataset and reproducibility in BioLab has been incorporated into the **5 Discussion and Future Work** section of the main text, highlighted in blue for emphasis.
2. A comprehensive definition of ambiguity is now included in the **3.3 Benchmark** section of the main document, also highlighted for easy identification.
3. Enhanced annotation details about procedure and cost have been integrated into the **3.2 Data annotation** section.
4. Webpage link (i.e. https://probio-dataset.github.io/) has been corrected in the main paper.

We appreciate all the suggestions made by reviewers to improve our work. It is our pleasure to hear your feedback, and we look forward to answering your follow-up questions.

ProBio Authors

---

### Decision · Program_Chairs · 2023-09-22

**Decision:**

Accept (Poster)

**Comment:**

The reviewers all agree that is is a strong dataset that holds value for the research community.  The dataset contributes to an area where there are few other public annotated data and labeling is time consuming/requires some skill.  While there was a concern about the quality of the annotations, it is clear that it will be useful to others. The authors have addressed the questions about the compensation, access to the dataset and how misuse will be mitigated.   Overall, the paper is well presented, detailed and has sufficient related work.

Strengths:
Useful, practical and valuable dataset for the research community
Clearly written and presented paper
Consideration of how the dataset can be released without misuse

Weaknesses:
Limited number of benchmark tasks
Further discussion of how the dataset will promote reproducibility would help